# Compositional Sculpting of Iterative Generative Processes

**Timur Garipov**[1][*]    **Sebastiaan De Peuter**[2]    **Ge Yang**[1,4]
**Vikas Garg**[2,5]    **Samuel Kaski**[2,3]    **Tommi Jaakkola**[1]

[1]MIT CSAIL    [2]Aalto University    [3]University of Manchester
[4]Institute for Artificial Intelligence and Fundamental Interactions    [5]YaiYai Ltd

## Abstract

High training costs of generative models and the need to fine-tune them for specific tasks have created a strong interest in model reuse and composition. A key challenge in composing iterative generative processes, such as GFlowNets and diffusion models, is that to realize the desired target distribution, all steps of the generative process need to be coordinated, and satisfy delicate balance conditions. In this work, we propose Compositional Sculpting: a general approach for defining compositions of iterative generative processes. We then introduce a method for sampling from these compositions built on classifier guidance. We showcase ways to accomplish compositional sculpting in both GFlowNets and diffusion models. We highlight two binary operations — the harmonic mean ($p_1 \otimes p_2$) and the contrast ($p_1 \, \unicode{x25D0} \, p_2$) between pairs, and the generalization of these operations to multiple component distributions. We offer empirical results on image and molecular generation tasks. Project codebase: https://github.com/timgaripov/compositional-sculpting.

## 1    Introduction

Large-scale general-purpose pre-training of machine learning models has produced impressive results in computer vision [1–3], image generation [4–6], natural language processing [7–11], robotics [12–14], and basic sciences [15]. By distilling vast amounts of data, such models can produce powerful inferences that lead to emergent capabilities beyond the specified training objective [16]. However, generic pre-trained models are often insufficient for specialized tasks in engineering and basic sciences. Field-adaptation via techniques such as explicit fine-tuning on bespoke datasets [17], human feedback [18], or cleverly designed prompts [19, 20] is therefore often required. Alternatively, capabilities of pre-trained models can be utilized and extended via model composition.

Compositional generation [21–28] views a complex target distribution in terms of simpler pre-trained building blocks which can be mixed and matched into a tailored solution to a specialized task. Given a set of base models capturing different properties of the data, composition provides a way to fuse these models into a single composite model with capacity greater than any individual base model. In this way it allows one to specify distributions over examples that exhibit multiple desired properties simultaneously [22]. The need to construct complex distributions adhering to multiple constraints arises in numerous practical multi-objective design problems such as molecule generation [29–31]. In this context, compositional modeling provides mechanisms for control of the resulting distribution and exploration of different trade-offs between the objectives and constraints.

Prior work on generative model composition [21, 22, 24] has developed operations for piecing together Energy-Based Models (EBMs) via algebraic manipulations of their energy functions. For example, consider two distributions $p_1(x) \propto \exp(-E_1(x))$ and $p_2(x) \propto \exp(-E_2(x))$ induced by energy

---

[*]Correspondence to Timur Garipov (`timur@csail.mit.edu`).

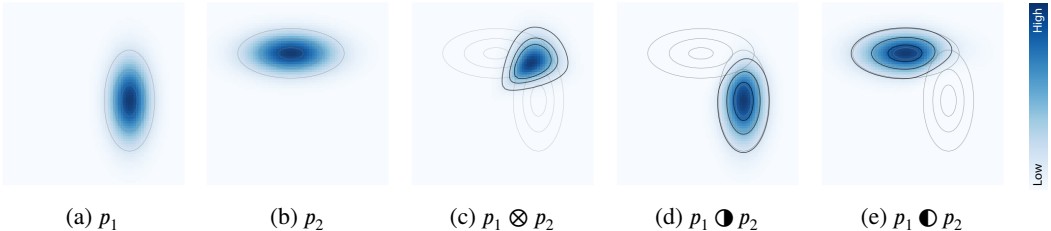

(a) $p_1$      (b) $p_2$      (c) $p_1 \otimes p_2$      (d) $p_1 \, \mathbb{O} \, p_2$      (e) $p_1 \, \mathbb{O} \, p_2$

Figure 1: **Composition operators.** (a,b) base distributions $p_1$ and $p_2$. (c) harmonic mean of $p_1$ and $p_2$. (d) contrast of $p_1$ with $p_2$ (e) reverse contrast $p_1 \, \mathbb{O} \, p_2$. Lines show the contours of the PDF level sets.

functions $E_1$ and $E_2$. Their *product* $p_{\text{prod}}(x) \propto p_1(x)p_2(x) \propto \exp(-(E_1(x) + E_2(x)))$ and *negation* $p_{\text{neg}}(x) \propto p_1(x)/(p_2(x))^\gamma \propto \exp(-(E_1(x) - \gamma E_2(x)))$, $\gamma > 0$ correspond to operations on the underlying energy functions. The product assigns high likelihood to points $x$ that have high likelihood under both base distributions but assigns low likelihood to points that have close-to-zero likelihood under one (or both). The negation distribution assigns high likelihood to points that are likely under $p_1$ but unlikely under $p_2$ and assigns low likelihood to points that are likely under $p_2$ but unlikely under $p_1$.

Iterative generative processes including diffusion models [5, 32–34] and GFlowNets [35, 36] progressively refine coarse objects into cleaner ones over multiple steps. Realizing effective compositions of these models is complicated by the fact that simple alterations in their generation processes result in non-trivial changes in the distributions of the final objects. Unlike for EBMs, products and negations of diffusion models cannot be realized through simple algebraic operations on their score functions. Du et al. [28] show that the result of the addition of score functions is not equal to the score of the diffused product distribution and develop a method that corrects the sum-of-scores sampling via additional MCMC steps nested under each step of the diffusion time loop.

Jain et al. [31] develop Multi-Objective GFlowNets (MOGFNs), an extension of GFlowNets for multi-objective optimization tasks. While a vanilla GFlowNet captures a distribution induced by a single reward (objective) function $p(x) \propto R(x)$ (see Section 2 for details), an MOGFN aims to learn a single conditional model that can realize distributions corresponding to various combinations (e.g. a convex combination) of multiple reward functions. Though a single MOGFN realizes a spectrum of compositions of base reward functions, the approach assumes access to the base rewards at training time. Moreover, MOFGNs require the set of possible composition operations to be specified at training time. In this work, we address post hoc composition of pre-trained GFlowNets (or diffusion models) and provide a way to create compositions that need not be specified in advance.

In this work, we introduce Compositional Sculpting, a general approach for the composition of pre-trained models. We highlight two special examples of binary operations — *harmonic mean*: $(p_1 \otimes p_2)$ and *contrast*: $(p_1 \, \mathbb{O} \, p_2)$. More general compositions are obtained as conditional distributions in a probabilistic model constructed on top of pre-trained base models. We show that these operations can be realized via classifier guidance. We provide results of empirical verification of our method on molecular generation (with GFlowNets) and image generation (with diffusion models).

## 2  Background

**Generative flow networks (GFlowNets).**  GFlowNets [35, 36] are an approach for generating structured objects (e.g. graphs) from a discrete space $\mathcal{X}$. Given a "reward function" $R(x) \geq 0$, a GFlowNet seeks to sample from $p(x) = R(x)/Z$, where $Z = \sum_x R(x)$, i.e. the model assigns larger probabilities to high-reward objects.

Starting at a fixed initial state $s_0$, objects $x$ are generated through a sequence of changes corresponding to a trajectory of incomplete states $\tau = (s_0 \to s_1 \to \ldots \to s_{n-1} \to x)$. The structure of possible trajectories corresponds to by a DAG $(\mathcal{S}, \mathcal{A})$ where $\mathcal{S}$ is a set of states (both complete and incomplete) and $\mathcal{A}$ is the set of directed edges (actions) $s \to s'$. The set of complete objects (terminal states) $\mathcal{X}$ is a subset of $\mathcal{S}$. The generation process starts at $s_0$ and follows a parameterized stochastic "forward policy" $P_F(s'|s; \theta)$ which for each state $s \in \mathcal{S} \setminus \mathcal{X}$ specifies a probability distribution over all possible successor states $s' : (s \to s') \in \mathcal{A}$. The process terminates once a terminal state is reached.

**Diffusion models.** Diffusion models [5, 32–34, 37] are a family of generative models developed for continuous domains. Given an empirical (data) distribution $\hat{p}(x) = 1/n \sum_i \delta_{\hat{x}_i}(x)$ in $\mathcal{X} = \mathbb{R}^d$, diffusion models seek to approximate $\hat{p}(x)$ via a generative process $p(x)$.

A diffusion process is a noising process that gradually destroys the original "clean" data $x$. Viewed as a stochastic differential equation (SDE) [34], it is a time-indexed collection of random variables $\{x_t\}_{t=0}^T$ in $\mathcal{X} = \mathbb{R}^d$ which interpolates between the data distribution $p_0(x) = \hat{p}(x)$ at $t = 0$ and a prior distribution $p_T(x)$ at $t = T$. The evolution of $x_t$ is described by the "forward SDE" $dx_t = f_t(x_t)\,dt + g_t\,dw_t$ with drift coefficient $f_t : \mathbb{R}^d \to \mathbb{R}^d$ and diffusion coefficient $g_t \in \mathbb{R}$. Here, $w_t$ is the standard Wiener process. Crucially, the coefficients $f_t$, $g_t$ are generally chosen such that the prior $p_T$ and the transition probabilities $p_{st}(x_t|x_s)$, $0 \le s < t \le T$ have a closed form (see [34]).

Song et al. [34] invoke a result from the theory of stochastic processes [38] which gives the expression for the reverse-time process or "backward SDE": $dx_t = \left[ f_t(x_t) - g_t^2 \nabla_x \log p_t(x_t) \right] dt + g_t\,d\overline{w}_t$, where $\overline{w}_t$ is the standard Wiener process in reversed time. This SDE involves the known coefficients $f_t$, $g_t$ and the unknown score function $\nabla_x \log p_t(\cdot)$ of the marginal distribution $p_t(\cdot)$ at time $t$. A "score network" $s_t(x; \theta)$ (a deep neural network with parameters $\theta$) is trained to approximate $\nabla_x \log p_t(x)$. Once $s_t(\cdot; \theta)$ is trained, sampling reduces to numerical integration of the backward SDE.

**Classifier guidance in diffusion models.** Classifier guidance [32, 39] is a technique for controllable generation in diffusion models. Suppose that each example $x_0$ is accompanied by a discrete class label $y$. The goal is to sample from the conditional distribution $p_0(x_0|y)$. The Bayes rule $p_t(x_t|y) \propto p_t(x_t)p_t(y|x_t)$ implies the score-function decomposition $\nabla_{x_t} \log p_t(x_t|y) = \nabla_{x_t} \log p_t(x_t) + \nabla_{x_t} \log p_t(y|x_t)$, where the first term is already approximated by a pre-trained unconditional diffusion model and the second term can be derived from a time-dependent classifier $p_t(y|x_t)$. Therefore, the stated goal can be achieved by first training the classifier $p_t(y|x_t)$ using noisy samples $x_t$ from the intermediate steps of the process, and then plugging in the expression for the conditional score into the backward SDE sampling process [34].

## 3 Related Work

**Generative model composition.** In Section 1 we reviewed prior work on energy-based composition operations and Multi-Objective GFlowNets. Learning mixtures of Generative Adversarial Networks has been addressed in [40], where the mixture components are learned simultaneously, and in [41], where the components are learned one by one in an adaptive boosting fashion. Algorithms for additive and multiplicative boosting of generative models have been developed in [42].

This work focuses on the composition of pre-trained models. Assuming that each pre-trained model represents the distribution of examples demonstrating certain concepts (e.g. molecular properties), the composition of models is equivalent to concept composition (e.g. property "A" AND property "B"). The inverse problem is known as "unsupervised concept discovery", where the goal is to automatically discover composable concepts from data. Unsupervised concept discovery and concept composition methods have been proposed for energy-based models [25] and text-to-image diffusion models [43].

**Controllable generation.** Generative model composition is a form of post-training control of the generation process – an established area of research. A simple approach to control is training a conditional generative model $p(x|c)$ on pairs $(x, c)$ of objects $x$ and conditioning information $c$. Annotations $c$ can be class labels [39], text prompts [4, 6, 44], semantic maps, and images [4]. Different from out work, this assumes that generation control operations are specified at training time. Dhariwal and Nichol [39] apply classifier guidance [32] on top of (conditional) diffusion models to improve their fidelity. Ho and Salimans [45] develop classifier-free guidance by combining conditional and unconditional score functions. In ControlNet [17], an additional network is trained to enable a pre-trained diffusion model to incorporate previously unavailable conditioning information. Meng et al. [46] and Couairon et al. [47] develop semantic image editing methods which first partially noise and then denoise an image to generate an edited version, possibly conditioned on a segmentation mask [47]. Similar to conditional diffusion models, conditional GFlowNets have been used to condition generation on reward exponents [36] or combinations of multiple predefined reward functions [31]. Note that the methods developed in this work can be combined with conditional diffusion models and GFlowNets: $p(x|c_1), \dots, p(x|c_m)$ can act as base generative models to be composed.

**Compositional generalization.** The notion of compositionality has a broad spectrum of inter-pretations across a variety of disciplines including linguistics, cognitive science, and philosophy. Hupkes et al. [48] collect a list of aspects of compositionality from linguistic and philosophical the-ories and design practical tests for neural language models. Conwell and Ullman [49] empirically examine the relational understanding of DALL-E 2 [50], a text-guided image generation model, and point out limitations in the model's ability to capture relations such as "in", "on", "hanging over", etc. In this work, we focus on a narrow but well-defined type of composition where we seek to algebraically compose probability densities in a controllable fashion, such that we can emphasize or de-emphasize regions in the data space where specific base distributions have high density.

**Connections between GFlowNets and diffusion models.** Our method is applicable to compositions of both GFlowNets and diffusion models. This is due to deep connections between these two model families. GFlowNets were initially developed for generating discrete (structured) data [36] and diffusion models were initially developed for continuous data [5, 32]. Lahlou et al. [51] develop an extension of GFlowNets for DAGs with continuous state-action spaces. Zhang et al. [52] point out unifying connections between GFlowNets and other generative model families, including diffusion models. In this work, we articulate another aspect of the relation between GFlowNets and diffusion models: in Section 5.2 we derive the expressions for mixture GFlowNet policies and classifier-guided GFlowNet policies analogous to those derived for diffusion models in [32, 39, 53, 54].

## 4 Compositional Sculpting of Generative Models

Suppose we can access a number of pre-trained generative models $\{p_i(x)\}_{i=1}^m$ over a common domain $\mathcal{X}$. We may wish to compose these distributions such that we can, say, draw samples that are likely to arise from $p_1(x)$ and $p_2(x)$, or that are likely to arise from $p_1(x)$ but not from $p_2(x)$. In other words, we wish to specify a distribution that we can shape to emphasize and de-emphasize specific base models.

### 4.1 Binary Composition Operations

Let us first focus on composing two base models. We could specify the composition as a weighted sum $\widetilde{p}(x) = \sum_{i=1}^2 \omega_i p_i(x)$ with weights $\omega_1, \omega_2 \geq 0$ summing to one. The weights determine the prevalence of each base model in the composition, but beyond that our control is limited. We cannot emphasize regions where $p_1$ and $p_2$ both have high density, or de-emphasize regions where $p_2$ has high density.

An alternative is to use conditioning to shape a prior $\widetilde{p}(x)$ based on the base models. When we condition $x$ on some observation $y_1$, the resulting posterior takes the form $\widetilde{p}(x|y_1) \propto \widetilde{p}(y_1|x)\widetilde{p}(x)$. Points $x$ that match $y_1$ according to $\widetilde{p}(y_1|x)$ will have increased density, and the density of points that do not match it decreases. Intuitively, by defining $y_1 \in \{1, 2\}$ as the event that $x$ was generated by a specific base model, we can shape a prior $\widetilde{p}(x)$ based on the densities of the base models. To this end we define a uniform prior over $y_k$ and define the conditional density $p(x|y_1 = i)$ to represent the fact that $x$ was generated from $p_i(x)$. This gives us the following model:

$$\widetilde{p}(x|y_1=i) = p_i(x), \quad \widetilde{p}(y_1=i) = 1/2, \quad \widetilde{p}(x) = \widetilde{p}(x|y_1=1)\widetilde{p}(y_1=1) + \widetilde{p}(x|y_1=2)\widetilde{p}(y_1=2). \quad (1)$$

Under this model, the prior $\widetilde{p}(x)$ is a uniform mixture of the base models. The likelihood of $y_1$

$$\widetilde{p}(y_1=1|x) = 1 - \widetilde{p}(y_1=2|x) = p_1(x)/(p_1(x) + p_2(x)), \quad (2)$$

implied by this model tells us how likely it is that $x$ was generated by $p_1(x)$ rather than $p_2(x)$. In fact, it corresponds to the output of an optimal classifier trained to tell $p_1(x)$ and $p_2(x)$ apart.

Our goal is to realize compositions which generate samples likely to arise from both $p_1(x)$ and $p_2(x)$ or from $p_1(x)$ but not $p_2(x)$. Thus we introduce a second observation $y_2 \in \{1, 2\}$ such that $y_1$ and $y_2$ are independent and identically distributed given $x$. The resulting model and inferred posterior are:

$$\widetilde{p}(x, y_1, y_2) = \widetilde{p}(x) \prod_{k=1}^2 \widetilde{p}(y_k|x), \quad \widetilde{p}(x) = \frac{1}{2}p_1(x) + \frac{1}{2}p_2(x), \quad \widetilde{p}(y_k=i|x) = p_i(x)/(p_1(x) + p_2(x)), \quad (3)$$

$$\widetilde{p}(x|y_1=i, y_2=j) \propto \widetilde{p}(x)\widetilde{p}(y_1=i|x)\widetilde{p}(y_2=j|x) \propto p_i(x)p_j(x)/(p_1(x) + p_2(x)). \quad (4)$$

The above posterior shows clearly how conditioning on observations $y_1 = i, y_2 = j$ has shaped the prior mixture to accentuate regions in the posterior where the observed base models $i, j$ have high density.

Conditioning on observations $y_1 = 1$ and $y_2 = 2$, or equivalently $y_1 = 2, y_2 = 1$, results in the posterior

$$(p_1 \otimes p_2)(x) := p(x|y_1 = 1, y_2 = 2) \propto p_1(x)p_2(x)/(p_1(x) + p_2(x)). \tag{5}$$

We refer to this posterior as the "**harmonic mean** of $p_1$ and $p_2$", and denote it as a binary operation $p_1 \otimes p_2$. Its value is high only at points that have high likelihood under both $p_1(x)$ and $p_2(x)$ at the same time (Figure 1(c)). Thus, the harmonic mean is an alternative to the product operation for EBMs. The harmonic mean is commutative ($p_1 \otimes p_2 = p_2 \otimes p_1$) and is undefined when $p_1$ and $p_2$ have disjoint supports, since then the RHS of (5) is zero everywhere.

Conditioning on observations $y_1 = 1$ and $y_2 = 1$ results in the posterior

$$(p_1 \mathbin{\text{\large◖}} p_2)(x) := \widetilde{p}(x|y_1 = 1, y_2 = 1) \propto (p_1(x))^2/(p_1(x) + p_2(x)). \tag{6}$$

We refer to this binary operation, providing an alternative to the negation operation in EBMs, as the "**contrast** of $p_1$ and $p_2$", and denote it as $p_1 \mathbin{\text{\large◖}} p_2$. The ratio (6) is high when $p_1(x)$ is high and $p_2(x)$ is low (Figure 1(d)). The contrast is not commutative ($p_1 \mathbin{\text{\large◖}} p_2 \neq p_2 \mathbin{\text{\large◖}} p_1$, unless $p_1 = p_2$). We denote the reverse contrast as $p_1 \mathbin{\text{\large◗}} p_2 = p_2 \mathbin{\text{\large◖}} p_1$. Appendix C provides a detailed comparison between the contrast and negation operations, and between the harmonic mean and product operations.

**Controlling the individual contributions of $p_1$ and $p_2$ to the composition.** In order to provide more control over the extent of individual contributions of $p_1$ and $p_2$ to the composition, we modify model (3). Specifically, we introduce an interpolation parameter $\alpha$ and change the likelihood of observation $y_2$ in (3): $\widetilde{p}(y_2 = i|x; \alpha) = (\alpha p_1(x))^{[i=1]} \cdot ((1 - \alpha)p_2(x))^{[i=2]} / (\alpha p_1(x) + (1 - \alpha)p_2(x))$, where $\alpha \in (0, 1)$ and $[\cdot]$ denotes the indicator function. Conditional distributions in this model give **harmonic interpolation**[2] and **parameterized contrast**:

$$(p_1 \otimes_{(1-\alpha)} p_2)(x) \propto \frac{p_1(x)p_2(x)}{\alpha p_1(x) + (1 - \alpha)p_2(x)}, \quad (p_1 \mathbin{\text{\large◖}}_{(1-\alpha)} p_2)(x) \propto \frac{(p_1(x))^2}{\alpha p_1(x) + (1 - \alpha)p_2(x)}. \tag{7}$$

**Operation chaining.** As the operations we have introduced result in proper distributions, we can create new $N$-ary operations by chaining binary (and $N$-ary) operations together. For instance, chaining binary harmonic means gives the harmonic mean of three distributions

$$((p_1 \otimes p_2) \otimes p_3)(x) = (p_1 \otimes (p_2 \otimes p_3))(x) \propto \frac{p_1(x)p_2(x)p_3(x)}{p_1(x)p_2(x) + p_1(x)p_3(x) + p_2(x)p_3(x)}. \tag{8}$$

## 4.2 Compositional Sculpting: General Approach

The above approach for realizing compositions of two base models can be generalized to compositions of $m$ base models $p_1(x), \ldots, p_m(x)$ controlled by $n$ observations. Though operator chaining can also realize compositions of $m$ base models, our generalized method allows us to specify compositions more flexibly and results in different compositions. We introduce an augmented probabilistic model $\widetilde{p}(x, y_1, \ldots, y_n)$ as a joint distribution over the original objects $x \in \mathcal{X}$ and $n$ observation variables $y_1 \in \mathcal{Y}, \ldots, y_n \in \mathcal{Y}$ where $\mathcal{Y} = \{1, \ldots, m\}$. By defining appropriate conditionals $p(y_k|x)$ we can controllably shape a prior $\widetilde{p}(x)$ into a posterior $\widetilde{p}(x|y_1, \ldots, y_n)$.

As in the binary case, we propose to use a uniformly-weighted mixture of the base models $\widetilde{p}(x) = (1/m) \cdot \sum_{i=1}^{m} p_i(x)$. The support of this mixture is the union of the supports of the base models: $\bigcup_{i=1}^{m} \text{supp}\{p_i(x)\} = \text{supp}\{\widetilde{p}(x)\}$. This is essential as the prior can only be shaped in places where it has non-zero density. As before we define the conditionals $p(y_k = i|x)$ to correspond to the observation that $x$ was generated by base model $i$. The resulting full model is

$$\widetilde{p}(x, y_1, \ldots, y_n) = \widetilde{p}(x) \prod_{k=1}^{n} \widetilde{p}(y_k|x), \ \widetilde{p}(x) = \frac{1}{m} \sum_{i=1}^{m} p_i(x), \ \widetilde{p}(y_k = i) = \frac{1}{m}, \ \widetilde{p}(y_k = i|x) = \frac{p_i(x)}{\sum_{j=1}^{m} p_j(x)}, \tag{9}$$

Note that under this model the mixture can be represented as $\widetilde{p}(x) = \sum_{y_k=1}^{m} \widetilde{p}(x|y_k)\widetilde{p}(y_k)$ for any $k$.

The inferred posterior over $x$ for this model is

$$\widetilde{p}(x|y_1 = i_1, \ldots, y_n = i_n) \propto \widetilde{p}(x)\widetilde{p}(y_1 = i_1, \ldots, y_n = i_n|x) \tag{10}$$

$$\propto \widetilde{p}(x) \prod_{k=1}^{n} \widetilde{p}(y_k = i_k|x) \propto \left( \prod_{k=1}^{n} p_{i_k}(x) \right) \Big/ \left( \sum_{j=1}^{m} p_j(x) \right)^{n-1}. \tag{11}$$

---

[2]the harmonic interpolation approaches $p_1$ when $\alpha \to 0$ and $p_2$ when $\alpha \to 1$

The posterior $\widetilde{p}(x|y_1 = i_1, \dots, y_n = i_n)$ is a composition of distributions $\{p_i(x)\}_{i=1}^m$ that can be adjusted by choosing values for $y_1, \dots, y_n$. By adding or omitting an observation $y_k = i$ we can *sculpt* the posterior to our liking, emphasizing or de-emphasizing regions of $\mathcal{X}$ where $p_i$ has high density. The observations can be introduced with multiplicities (e.g., $y_1 = 1, y_2 = 1, y_3 = 2$) to further strengthen the effect. Moreover, one can choose to introduce all observations simultaneously as in (10) or sequentially as in (11). As we show below (Section 5.1 for GFlowNets; Appendix A.2 for diffusion models), the composition (10) can be realized by a sampling policy that can be expressed as a function of the pre-trained (base) sampling policies.

**Special instances and general formulation.** The general approach outlined in this section is not limited to choices we made to construct the model in equation (9), i.e. $\widetilde{p}(x)$ does not have to be a uniformly weighted mixture of the base distributions, $y_1, \dots, y_n$ do not have to be independent and identically distributed given $x$, and different choices of the likelihood $\widetilde{p}(y = i|x)$ are possible. For instance, in the model for parameterized operations (7) the likelihoods of observations $\widetilde{p}(y_1|x), \widetilde{p}(y_2|x)$ differ.

# 5 Compositional Sculpting of Iterative Generative Processes

In this Section, we show how to apply the model above to compose GFlowNets, and how one can use classifier guidance to sample from the composition. The similar method for diffusion model composition is described in Appendix A.2.

## 5.1 Composition of GFlowNets

Besides a sample $x$ from $p_i(x)$, a GFlowNet also generates a trajectory $\tau$ which ends in the state $x$. Thus, we extend the model $\widetilde{p}(x, y_1, \dots, y_n)$, described above, and introduce $\tau$ as a variable with conditional distribution $\widetilde{p}(\tau|y_k = i) = \prod_{t=0}^{|\tau|-1} p_{i,F}(s_{t+1}|s_t)$, where $p_{i,F}$ is the forward policy of the GFlowNet that samples from $p_i$.

Our approach for sampling from the composition is conceptually simple. Given $m$ base GFlowNets that sample from $p_1, \dots, p_m$ respectively, we start by defining the prior $\widetilde{p}(x)$ as the uniform mixture of these GFlowNets. Proposition 5.1 shows that this mixture can be realized by a policy constructed from the forward policies of the base GFlowNets. We then apply classifier guidance to this mixture to sample from the composition. Proposition 5.2 shows that classifier guidance results in a new policy which can be constructed directly from the GFlowNet being guided.

**Proposition 5.1** (GFlowNet mixture policy). *Suppose distributions $p_1(x), \dots, p_m(x)$ are realized by GFlowNets with forward policies $p_{1,F}(\cdot|\cdot), \dots, p_{m,F}(\cdot|\cdot)$. Then, the mixture distribution $p_M(x) = \sum_{i=1}^m \omega_i p_i(x)$ with $\omega_1, \dots, \omega_m \geq 0$ and $\sum_{i=1}^m \omega_i = 1$ is realized by the GFlowNet forward policy*

$$p_{M,F}(s'|s) = \sum_{i=1}^m p(y = i|s)p_{i,F}(s'|s), \tag{12}$$

*where $y$ is a random variable such that the joint distribution of a GFlowNet trajectory $\tau$ and $y$ is given by $p(\tau, y = i) = \omega_i p_i(\tau)$ for $i \in \{1, \dots, m\}$.*

**Proposition 5.2** (GFlowNet classifier guidance). *Consider a joint distribution $p(x, y)$ over a discrete space $\mathcal{X} \times \mathcal{Y}$ such that the marginal $p(x)$ is realized by a GFlowNet with forward policy $p_F(\cdot|\cdot)$. Assume that the joint distribution of $x$, $y$, and GFlowNet trajectories $\tau = (s_0 \dots s_n = x)$ decomposes as $p(\tau, x, y) = p(\tau, x)p(y|x)$, i.e. $y$ is independent of the intermediate states $\{s_i\}_{i=0}^{n-1}$ in $\tau$ given $x$. Then,*

*1. For all non-terminal nodes $s \in S \setminus \mathcal{X}$, the probabilities $p(y|s)$ satisfy*

$$p(y|s) = \sum_{s':(s \to s') \in \mathcal{A}} p_F(s'|s)p(y|s'). \tag{13}$$

*2. The conditional distribution $p(x|y)$ is realized by the classifier-guided policy*

$$p_F(s'|s, y) = p_F(s'|s) \cdot p(y|s') / p(y|s). \tag{14}$$

Note that (13) ensures that $\sum_{s':(s \to s') \in \mathcal{A}} p_F(s'|s, y) = 1$.

Proposition 5.1 is analogous to results on mixtures of diffusion models (Theorem 1 of Peluchetti [53], Theorem 1 of Lipman et al. [54]). Proposition 5.2 is analogous to classifier guidance for diffusion models [32, 39]. To the best of our knowledge, our work is the first to derive both results for GFlowNets.

Theorem 5.3 summarizes our approach. The propositions and the theorem are proved in Appendix D.

**Theorem 5.3.** *Suppose distributions $p_1(x), \ldots, p_m(x)$ are realized by GFlowNets with forward policies $p_{1,F}(\cdot|\cdot), \ldots, p_{m,F}(\cdot|\cdot)$ respectively. Let $y_1, \ldots, y_n$ be random variables defined by* (9)*. Then, the conditional $\widetilde{p}(x|y_1, \ldots, y_n)$ is realized by the forward policy*

$$p_F(s'|s, y_1, \ldots, y_n) = \frac{\widetilde{p}(y_1, \ldots, y_n|s')}{\widetilde{p}(y_1, \ldots, y_n|s)} \sum_{i=1}^{m} p_{i,F}(s'|s)\widetilde{p}(y=i|s) \tag{15}$$

Note that the result of conditioning on observations $y_1, \ldots, y_n$ is just another GFlowNet policy. Therefore, to condition on more observations, one can apply classifier guidance repeatedly.

## 5.2 Classifier Training (GFlowNets)

The evaluation of policy (15) requires knowledge of the probabilities $\widetilde{p}(y_1, \ldots, y_n|s)$, $\widetilde{p}(y|s)$. These probabilities can be estimated by a classifier fitted to trajectories sampled from the base GFlowNets.

Let $\widetilde{Q}_\phi(y_1, \ldots, y_n|s)$ be a classifier with parameters $\phi$ that we wish to train to approximate the ground-truth conditional $\widetilde{p}(y_1, \ldots, y_n|s)$. Under model (9), $y_1, \ldots, y_n$ are dependent given a state $s \in \mathcal{S} \setminus \mathcal{X}$, but, are independent given a terminal state $x \in \mathcal{X}$. This motivates separate treatment of terminal and non-terminal states.

**Learning the terminal state classifier.** For a terminal state $x$, the variables $y_1, \ldots, y_n$ are independent, hence we can use the factorization $\widetilde{Q}_\phi(y_1, \ldots, y_n|x) = \prod_{k=1}^{n} \widetilde{Q}_\phi(y_k|x)$. Moreover, all distributions on the *r.h.s.* must be the same, i.e. it is enough to learn just $\widetilde{Q}_\phi(y_1|x)$. This marginal classifier can be learned by minimizing the cross-entropy loss

$$\mathcal{L}_T(\phi) = \mathop{\mathbb{E}}_{(\widehat{x},\widehat{y}_1)\sim\widetilde{p}(y_1)\widetilde{p}(x|y_1)} \left[ -\log \widetilde{Q}_\phi(y_1=\widehat{y}_1|x=\widehat{x}) \right]. \tag{16}$$

Here $\widehat{y}_1$ is sampled from $\widetilde{p}(y_1)$, which is uniform under our choice of $\widetilde{p}(x)$. Then, $\widehat{x}|(y_1=\widehat{y}_1)$ is generated from the base GFlowNet $p_{\widehat{y}_1}$, since (9) implies that $\widetilde{p}(x|y=\widehat{y}_1) = p_{\widehat{y}_1}(x)$.

**Learning the non-terminal state classifier.** Given a non-terminal state $s \in \mathcal{S} \setminus \mathcal{X}$, we need to model $y_1, \ldots, y_n$ jointly, and training requires sampling tuples $(\widehat{s}, \widehat{y}_1, \ldots, \widehat{y}_n)$. Non-terminal states $s$ can be generated as intermediate states in trajectories $\tau = (s_0 \to s_1 \to \ldots \to x)$. Given a sampled trajectory $\widehat{\tau}$ and a set of labels $\widehat{y}_1, \ldots, \widehat{y}_n$ we denote the trajectory cross-entropy by

$$\ell(\widehat{\tau}, \widehat{y}_1, \ldots, \widehat{y}_n; \phi) = \sum_{t=0}^{|\tau|-1} \left[ -\log \widetilde{Q}_\phi(y_1=\widehat{y}_1, \ldots, y_n=\widehat{y}_n|s=\widehat{s}_t) \right]. \tag{17}$$

Pairs $(\widehat{\tau}, \widehat{y}_1)$ can be generated in the same way as in the terminal classifier training above: 1) $\widehat{y}_1 \sim \widetilde{p}(y_1)$; 2) $\widehat{\tau} \sim p_{\widehat{y}_1}(\tau)$. Sampling $\widehat{y}_2, \ldots, \widehat{y}_n$ given $\widehat{x}$ (the terminal state of $\widehat{\tau}$) requires access to values $\widetilde{p}(y_k = \widehat{y}_k|\widehat{x})$, which are not directly available. However, if the terminal classifier is learned as described above, the estimates $w_i(\widehat{x}; \phi) = \widetilde{Q}_\phi(y_1 = i|x = \widehat{x})$ can be used instead. In this case, the loss and the sampling procedure for the non-terminal classifier rely on the outputs of the terminal classifier. In order to train two classifiers simultaneously, and avoid the instability due to a feedback loop, we employ the "target network" technique developed in the context of deep Q-learning [55]. We introduce a "target network" parameter vector $\overline{\phi}$ which is used to produce the estimates $w_i(\widehat{x}; \overline{\phi})$ for the non-terminal loss. We update $\overline{\phi}$ as the exponential moving average of the recent iterates of $\phi$.

After putting all components together the training loss for the non-terminal state classifier is

$$\mathcal{L}_N(\phi, \overline{\phi}) = \mathop{\mathbb{E}}_{(\widehat{\tau},\widehat{y}_1)\sim\widetilde{p}(\tau,y_1)} \left[ \sum_{\widehat{y}_2=1}^{m} \cdots \sum_{\widehat{y}_n=1}^{m} \left( \prod_{k=2}^{n} w_{\widehat{y}_k}(\widehat{x}; \overline{\phi}) \right) \ell(\widehat{\tau}, \widehat{y}_1, \ldots, \widehat{y}_n; \phi) \right]. \tag{18}$$

We refer the reader to Appendix D.4 for a more detailed derivation of the loss (18). Algorithm A.1 shows the complete classifier training procedure.

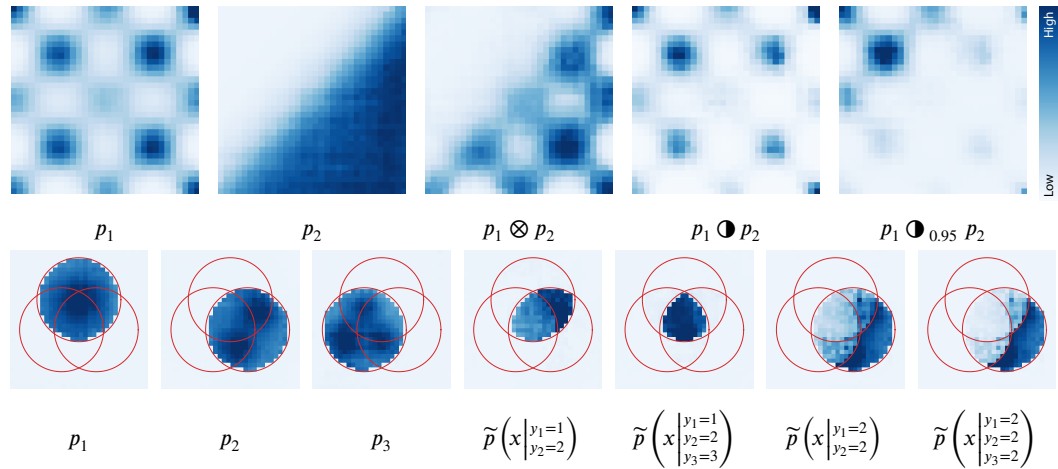

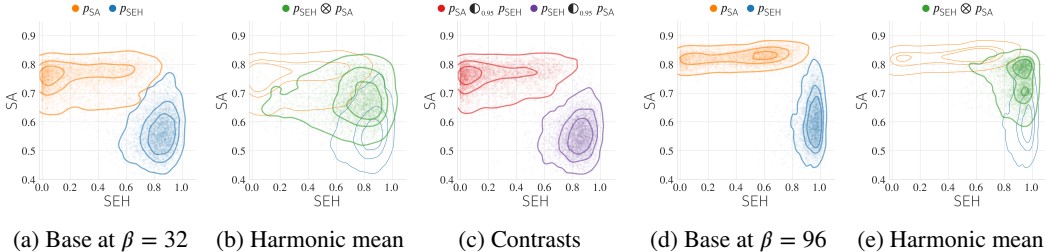

Figure 2: **Composed GFlowNets on** $32 \times 32$ **grid domain.** (Top) operations on two distributions. (Bottom) operations on three distributions. Cell probabilities are shown with color, darker is higher. Red circles indicate the high probability regions of $p_1, p_2, p_3$.

|     |     |     |     |     |
| --- | --- | --- | --- | --- |
| (a) Base at $\beta = 32$ | (b) Harmonic mean | (c) Contrasts | (d) Base at $\beta = 96$ | (e) Harmonic mean |

Figure 3: **Reward distributions in the molecular generation domain.** (a) Base GFlowNets at $\beta=32$: $p_{\text{SEH}}$ and $p_{\text{SA}}$ are trained with $R_{\text{SEH}}(x)^{32}$ and $R_{\text{SA}}(x)^{32}$. (b) harmonic mean of $p_{\text{SEH}}$ and $p_{\text{SA}}$ at $\beta=32$. (c) contrasts at $\beta=32$. (d) base GFlowNets at $\beta=96$. (e) harmonic mean at $\beta=96$. Lines show the contours of the level sets of the kernel density estimates in the $(R_{\text{SEH}}, R_{\text{SA}})$ plane.

## 6 Experiments

**2D distributions via GFlowNet.** We tested our GFlowNet composition method on 2D grid [36], a controlled domain, where the ground-truth composite distributions can be evaluated directly.

In the 2D grid domain, the states are the cells of an $H \times H$ grid. The starting state is the upper-left cell $s_0 = (0, 0)$. At each state, the allowed actions are: 1) move right; 2) move down; 3) terminate the trajectory at the current position. We first trained GFlowNets $p_i(x) \propto R_i(x)$ with reward functions $R_i(x) > 0$, and then trained classifiers and constructed compositions following Theorem 5.3.

Figure 2 (top row) shows the distributions obtained by composing two pre-trained GFlowNets (top row; left). The harmonic mean $p_1 \otimes p_2$, covers the regions that have high probability under both $p_1$ and $p_2$ and excludes locations where either of the probabilities is low. $p_1 \oplus p_2$ resembles $p_1$ but the relative masses of the modes of $p_1$ are modulated by $p_2$: regions with high $p_2$ have lower probability under contrast. The parameterized contrast $p_1 \oplus_{0.95} p_2$ with $\alpha = 0.05$ magnifies the contrasting effect: high $p_2(x)$ implies very low $(p_1 \oplus_{0.95} p_2)(x)$. The bottom row of Figure 2 shows operations on 3 distributions. The conditional $\widetilde{p}(x|y_1{=}1, y_2{=}2)$ is concentrated on the points that have high likelihood under both $p_1$ and $p_2$. Similarly, the value $\widetilde{p}(x|y_1{=}1, y_2{=}2, y_3{=}3)$ is high if $x$ is likely to be observed under all three distributions at the same time. The conditionals $\widetilde{p}(x|y_1{=}2, y_2{=}2)$ and $\widetilde{p}(x|y_1{=}2, y_2{=}2, y_3{=}2)$ highlight the points with high $p_2(x)$ but low $p_1(x)$ and $p_3(x)$. Conditioning on three labels results in a sharper distribution compared to double-conditioning. We provide quantitative results and further details in Appendix F.1. The classifier learning curves are provided in Appendix G.4.

**Molecule generation via GFlowNet.** Next, we evaluated our method for GFlowNet composition on a large and highly structured data space, and assessed the effect that composition operations have

Table 1: Reward distributions of composite GFlowNets.

| | SEH | low | | | | high | | | |
| | SA | low | | high | | low | | high | |
| | QED | low | high | low | high | low | high | low | high |
|---|---|---|---|---|---|---|---|---|---|
| $p_{SEH}$ | | 0 | 0 | 0 | 0 | 62 | 9 | 24 | 5 |
| $p_{SA}$ | | 0 | 0 | 73 | 4 | 0 | 0 | 18 | 5 |
| $p_{QED}$ | | 0 | 40 | 0 | 26 | 0 | 21 | 0 | 13 |
| (a) $y=\{\text{SEH, SA}\}$ | | 1 | 0 | 16 | 2 | 6 | 3 | 54 | 18 |
| (b) $y=\{\text{SEH, QED}\}$ | | 0 | 11 | 0 | 4 | 1 | 48 | 4 | 32 |
| (c) $y=\{\text{SA, QED}\}$ | | 0 | 15 | 1 | 42 | 0 | 8 | 2 | 32 |
| (d) $y=\{\text{SEH, SA, QED}\}$ | | 0 | 7 | 2 | 11 | 2 | 19 | 10 | 49 |
| (e) $y=\{\text{SEH, SEH, SEH}\}$ | | 0 | 0 | 0 | 0 | 63 | 9 | 24 | 4 |
| (f) $y=\{\text{SA, SA, SA}\}$ | | 0 | 0 | 74 | 5 | 0 | 0 | 17 | 4 |
| (g) $y=\{\text{QED, QED, QED}\}$ | | 0 | 40 | 0 | 23 | 0 | 23 | 0 | 14 |

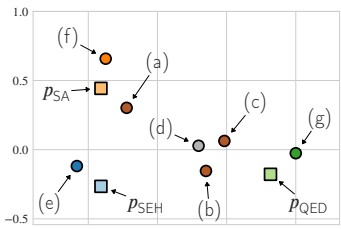

Figure 4: 2D t-SNE embeddings of three base GFlowNets trained with $R_{SEH}(x)^\beta$, $R_{SA}(x)^\beta$, $R_{QED}(x)^\beta$ at $\beta = 32$ and their compositions. The t-SNE embeddings are computed based on pairwise earth mover's distances between the distributions. Labels (a)-(g) match rows in Table 1.

In each row, the numbers show the percentage of the samples from the respective model that fall into one of 8 bins according to rewards. The "low" and "high" categories are decided by thresholding SEH: 0.5, SA: 0.6, QED: 0.25.

on resulting data distributions in a practical setting. To that end, we conducted experiments with GFlowNets trained for the molecular generation task proposed by Bengio et al. [36]. We train base GFlowNets with 3 reward functions $R_{SEH}(x)$, $R_{SA}(x)$, and $R_{QED}(x)$ measuring 3 distinct molecular properties (details in Appendix F.2). Following Bengio et al. [36], we introduced the parameter $\beta$ which controls the sharpness (temperature) of the target distribution: $p(x) \propto R(x)^\beta$, increasing $\beta$ results in a distribution skewed towards high-reward objects. We experimented with $\beta = 32$ and $\beta = 96$ (Figures 3(a),3(d)). After training base GFlowNets with the respective reward functions, we trained classifiers with Algorithm A.1. In order to evaluate the base models and the composed GFlowNet policies, we generated 5 000 samples with each policy and anaylzed the samples. Further details are in Appendix F.2. Classifier learning curves are provided in Appendix G.4. Sample diversity statistics of base GFlowNets at different values of $\beta$ are provided in Appendix G.5.

Figure 3 shows reward distributions of base GFlowNets (trained with rewards $R_{SEH}(x)^\beta$, $R_{SA}(x)^\beta$, $\beta \in \{32, 96\}$) and their compositions. Base GFlowNet distributions are concentrated on examples that score high in their respective rewards. For each model, there is considerable variation in the reward that was not used for training. The harmonic mean operation (Figures 3(b), 3(e)) results in distributions that are concentrated on the samples scoring high in both rewards. The contrast operation (Figure 3(c)) has the opposite effect: the distributions are skewed towards the examples scoring high in only one of the original rewards. Note that the tails of the contrast distributions are retreating from the area covered by the harmonic mean.

We show reward distribution statistics of three GFlowNets (trained with SEH, SA, and QED at $\beta = 32$) and their compositions in Table 1. Each row of the table gives a percentage breakdown of the samples from a given model into one of $2^3 = 8$ bins according to rewards. For all three base models, the majority of the samples fall into the "high" category according to the respective reward, while the rewards that were not used for training show variation. Conditioning on two different labels (e.g. $y=\{\text{SEH, QED}\}$) results in concentration on examples that score high in two selected rewards, but not necessarily scoring high in the reward that was not selected. The conditional $y=\{\text{SEH, QED, SA}\}$ shifts the focus to examples that have all three properties.

Figure 4 shows 2D embeddings of the distributions appearing in Table 1. The embeddings were computed with t-SNE based on the pairwise earth mover's distances (details in Appendix F.2, complete summary of distribution distances in Table G.5). The configuration of the embeddings provides insight into the relative positions of the base models and conditionals in the distribution space. The points corresponding to the pairwise conditionals lie in between the two base models selected for conditioning. The conditional $y=\{\text{SEH, SA, QED}\}$ appears to be near the centroid of the triangle $(p_{SEH}, p_{SA}, p_{QED})$ and lies close the the pairwise conditionals. The distributions obtained by repeated conditioning on the same label (e.g. $y=\{\text{SEH, SEH, SEH}\}$) are spread out to the boundary, lying closer to the respective base distributions and relatively far from pairwise conditionals.

**Colored MNIST generation via diffusion models.** Finally, we empirically tested our method for the composition of diffusion models on an image generation task. In this experiment, we composed

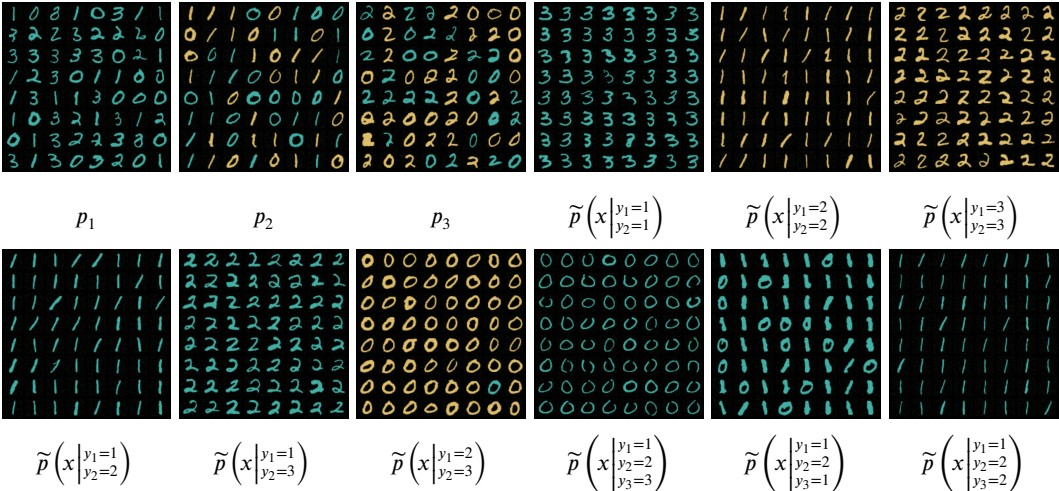

Figure 5: **Composed diffusion models on colored MNIST.** Samples from 3 pre-trained diffusion models and their various compositions.

three diffusion models trained to generate MNIST [56] digits {0, 1, 2, 3} in two colors: cyan and beige. Each model was trained to generate digits with a specific property: $p_1$ generated cyan digits, $p_2$ generated digits less than 2, and $p_3$ generated even digits.

We built the composition iteratively via the factorization $\widetilde{p}(x|y_1, y_2, y_3) \propto \widetilde{p}(x)\widetilde{p}(y_1, y_2|x)\widetilde{p}(y_3|x, y_1, y_2)$. To this end, we first trained a classifier $\widetilde{Q}(y_1, y_2|x_t)$ on trajectories sampled from the base models. This allowed us to generate samples from $\widetilde{p}(x|y_1, y_2)$. We then trained an additional classifier $\widetilde{Q}(y_3|x_t, y_1, y_2)$ on trajectories from compositions defined by $(y_1, y_2)$ to allow us to sample from $\widetilde{p}(x|y_1, y_2, y_3)$. Further details can be found in Appendix F.3.

Figure 5 shows samples from the pre-trained models and from selected compositions. The negating effect of *not* conditioning on observations is clearly visible in the compositions using two variables. For example, $\widetilde{p}(x|y_1 = 1, y_2 = 1)$ only generates cyan 3's. Because there we *do not* condition on $p_2$ or $p_3$, the composition excludes digits that have high probability under $p_2$ or $p_3$, i.e. those that are less than 2 or even. In $\widetilde{p}(x|y_1 = 1, y_2 = 3)$, cyan even digits have high density under both $p_1$ and $p_3$, but because $p_2$ is not conditioned on, the composition excludes digits less than two (i.e. cyan 0's). Finally, $\widetilde{p}(x|y_1 = 1, y_2 = 2, y_3 = 3)$ generates only cyan 0's, on which all base models have high density.

# 7 Conclusion

We introduced Compositional Sculpting, a general approach for composing iterative generative models. Compositions are defined through "observations", which enable us to emphasize or de-emphasize the density of the composition in regions where specific base models have high density. We highlighted two binary compositions, harmonic mean and contrast, which are analogous to the product and negation operations defined on EBMs. A crucial feature of the compositions we have introduced is that we can sample from them directly. By extending classifier guidance we are able to leverage the generative capabilities of the base models to produce samples from the composition. Through empirical experiments, we validated our approach for composing diffusion models and GFlowNets on toy domains, molecular generation, and image generation.

**Broader impact.** We proposed a mathematical framework and methods for the composition of pre-trained generative models. While the primary emphasis of our work is on advancing foundational research on generative modeling methodology and principled sampling techniques, our work inherits ethical concerns associated with generative models such as creation of deepfake content and mis-information dissemination, as well as reproduction of biases present in the datasets used for model training. If not carefully managed, these models can perpetuate societal biases, exacerbating issues of fairness and equity. Our work further contributes to research on the reuse of pre-trained models. This research direction promotes eco-friendly AI development, with the long-term goal of reducing energy consumption and carbon emissions associated with large-scale generative model training.

## Acknowledgements

TG and TJ acknowledge support from the Machine Learning for Pharmaceutical Discovery and Synthesis (MLPDS) consortium, DARPA Accelerated Molecular Discovery program, the NSF Expeditions grant (award 1918839) "Understanding the World Through Code", and from the MIT-DSTA collaboration.

SK and SDP were supported by the Technology Industries of Finland Centennial Foundation and the Jane and Aatos Erkko Foundation under project Interactive Artificial Intelligence for Driving R&D, the Academy of Finland (flagship programme: Finnish Center for Artificial Intelligence, FCAI; grants 328400, 345604 and 341763), and the UKRI Turing AI World-Leading Researcher Fellowship, EP/W002973/1.

VG acknowledges support from the Academy of Finland (grant decision 342077) for "Human-steered next-generation machine learning for reviving drug design", the Saab-WASP initiative (grant 411025), and the Jane and Aatos Erkko Foundation (grant 7001703) for "Biodesign: Use of artificial intelligence in enzyme design for synthetic biology".

GY acknowledges support from the National Science Foundation under Cooperative Agreement PHY-2019786 (The NSF AI Institute for Artificial Intelligence and Fundamental Interactions, http://iaifi.org).

We thank Sammie Katt and Pavel Izmailov for the helpful discussions and assistance in making the figures.

We thank NeurIPS 2023 anonymous reviewers for the helpful feedback on our work.

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

# A    Compositional Sculpting of Iterative Generative Processes (Section 5, continued)

## A.1    Classifier Training (GFlowNets): Algorithm

---
**Algorithm A.1** Compositional Sculpting: classifier training

---
1: Initialize $\phi$ and set $\overline{\phi} = \phi$
2: **for** step $= 1, \dots,$ num_steps **do**
3:    **for** $i = 1, \dots, m$ **do**
4:       Sample $\hat{\tau}_i \sim p_i(\tau)$
5:    **end for**
6:    $\mathcal{L}_T(\phi) = -\sum_{i=1}^{m} \log \widetilde{Q}_\phi(y_1 = i | x = \hat{x}_i)$    {Terminal state loss (16)}
7:    $w_i(\hat{x}_j; \overline{\phi}) = \widetilde{Q}_{\overline{\phi}}(y_k = i | x = \hat{x}_j), \ i, j \in \{1, \dots m\}$    {Terminal probability estimates}

   {Non-terminal state loss (17)-(18)}
8:    $\mathcal{L}_N(\phi, \overline{\phi}) = \sum_{\hat{y}_1=1}^{m} \dots \sum_{\hat{y}_n=1}^{m} \left( \prod_{k=2}^{n} w_{\hat{y}_k}(\hat{x}_{\hat{y}_1}; \overline{\phi}) \right) \ell(\hat{\tau}_{\hat{y}_1}, \hat{y}_1, \dots \hat{y}_n; \phi)$
9:    $\mathcal{L}(\phi, \overline{\phi}) = \mathcal{L}_T(\phi) + \gamma(\text{step}) \cdot \mathcal{L}_N(\phi, \overline{\phi})$
10:    Update $\phi$ using $\nabla_\phi \mathcal{L}(\phi, \overline{\phi})$; update $\overline{\phi} = \beta\overline{\phi} + (1 - \beta)\phi$
11: **end for**

---

## A.2    Composition of Diffusion Models

In this section, we show how the method introduced above can be applied to diffusion models. First, we adapt the model we introduced in (9)-(11) to diffusion models. A diffusion model trained to sample from $p_i(x)$ generates a trajectory $\tau = \{x_t\}_{t=0}^{T}$ over a range of time steps which starts with a randomly sampled state $x_T$ and ends in $x_0$, where $x_0$ has distribution $p_{i,t=0}(x) = p_i(x)$. Thus, we must adapt our model to reflect this. We introduce a set of mutually dependent variables $x_t$ for $t \in (0, T]$ with as conditional distribution the transition kernel of the diffusion model $p_i(x_t | x_0)$.

Given $m$ base diffusion models that sample from $p_1, \dots, p_m$ respectively, we define the prior $\widetilde{p}(x)$ as a mixture of these diffusion models. Proposition A.1 shows that this mixture is a diffusion model that can be constructed directly from the base diffusion models. We then apply classifier guidance to this mixture to sample from the composition. We present an informal version of the proposition below. The required assumptions and the proof are provided in Appendix D.5.

**Proposition A.1** (Diffusion mixture SDE). *Suppose distributions $p_1(x), \dots, p_m(x)$ are realized by diffusion models with forward SDEs $dx_{i,t} = f_{i,t}(x_{i,t}) dt + g_{i,t} dw_{i,t}$ and score functions $s_{i,t}(\cdot)$, respectively. Then, the mixture distribution $p_M(x) = \sum_{i=1}^{m} \omega_i p_i(x)$ with $\omega_1 \dots \omega_m \geq 0$ and $\sum_{i=1}^{m} \omega_i = 1$ is realized by a diffusion model with forward SDE*

$$dx_t = \underbrace{\left[ \sum_{i=1}^{m} p(y = i | x_t) f_{i,t}(x_t) \right]}_{f_{M,t}(x_t)} dt + \underbrace{\sqrt{\sum_{i=1}^{m} p(y = i | x_t) g_{i,t}^2}}_{g_{M,t}(x_t)} dw_t, \tag{19}$$

*and backward SDE*

$$dx_t = \left[ \sum_{i=1}^{m} p(y = i | x_t) \left( f_{i,t}(x_t) - g_{i,t}^2 s_{i,t}(x_t) \right) \right] dt + \sqrt{\sum_{i=1}^{m} p(y = i | x_t) g_{i,t}^2} \, d\overline{w}_t, \tag{20}$$

*with*

$$p(y = i | x_t) = \frac{\omega_i p_{i,t}(x_t)}{\sum_{j=1}^{m} \omega_j p_{j,t}(x_t)}. \tag{21}$$

If the base diffusion models have a common forward SDE $dx_{i,t} = f_t(x_{i,t}) dt + g_t dw_{i,t}$, equations (19)-(20) simplify to

$$dx_t = f_t(x_t)dt + g_t dw_t, \quad dx_t = \left[ f_t(x_t) - g_t^2 \left( \sum_{i=1}^{m} p(y=i|x_t)s_{i,t}(x_t) \right) \right] dt + g_t d\overline{w}_t. \quad (22)$$

Theorem A.2 summarizes the overall approach.

**Theorem A.2.** *Suppose distributions $p_1(x), \ldots, p_m(x)$ are realized by diffusion models with forward SDEs $dx_{i,t} = f_{i,t}(x_{i,t}) dt + g_{i,t} dw_{i,t}$ and score functions $s_{i,t}(\cdot)$, respectively. Let $y_1, \ldots y_n$ be random variables defined by (9). Then, the conditional $\widetilde{p}(x|y_1, \ldots, y_n)$ is realized by a classifier-guided diffusion with backward SDE*

$$dx_t = v_{C,t}(x_t, y_1, \ldots, y_n)dt + g_{C,t}(x_t)d\overline{w}_t, \quad (23)$$

*with*

$$v_{C,t}(x_t, y_1, \ldots, y_n) = \sum_{i=1}^{m} \widetilde{p}(y=i|x_t) \Big( f_{i,t}(x_t) - g_{i,t}^2 \Big( s_{i,t}(x_t) + \nabla_{x_t} \log \widetilde{p}(y_1, \ldots, y_n|x_t) \Big) \Big), \quad (24)$$

$$g_{C,t}(x_t) = \sqrt{ \sum_{i=1}^{m} \widetilde{p}(y=i|x_t) g_{i,t}^2 }. \quad (25)$$

The proof of Theorem A.2 is provided in Appendix D.6.

## A.3 Classifier Training (Diffusion Models)

We approximate the inferential distributions in equations (22) and (23) with a time-conditioned classifier $\widetilde{Q}_\phi(y_1, \ldots, y_n|x_t)$ with parameters $\phi$. Contrary to GFlowNets, which employed a terminal and non-terminal state classifier, here we only need a single time-dependent classifier. The classifier is trained with different objectives on terminal and non-terminal states. The variables $y_1, \ldots, y_n$ are dependent given a state $x_t$ for $t \in [0, T]$, but are independent given the terminal state $x_T$. Thus, when training on terminal states we can exploit this independence. Furthermore, we generally found it beneficial to initially train only on terminal states. The loss for the non-terminal states depends on classifications of the terminal state of the associated trajectories, thus by minimizing the classification error of terminal states first, we reduce noise in the loss calculated for the non-terminal states later.

For a terminal state $x_0$, the classifier $\widetilde{Q}_\phi(y_1, \ldots, y_n|x_t)$ can be factorized as $\prod_{k=1}^{n} \widetilde{Q}_\phi(y_k|x_0)$. Hence we can train $\widetilde{Q}$ by minimizing the cross-entropy loss

$$\mathcal{L}_T(\phi) = \mathbb{E}_{(\widehat{x}_0, \widehat{y}_1) \sim \widetilde{p}(x, y_1)} \left[ -\log \widetilde{Q}_\phi(y_1 = \widehat{y}_1|x_0 = \widehat{x}_0) \right]. \quad (26)$$

Samples $\widetilde{p}(x_0, y_1)$ can be generated according to the factorization $\widetilde{p}(y_1)\widetilde{p}(x_0|y_1)$. First, $\widehat{y}_1$ is sampled from $\widetilde{p}(y_1)$, which is uniform under our choice of $\widetilde{p}(x_0)$. Then, $\widehat{x}_0|(y_1 = \widehat{y}_1)$ is generated from the reverse SDE of base diffusion model $p_{\widehat{y}_1}(x)$. Note that equation (9) implies that all observations have the same conditional distribution given $x$. Thus, $\widetilde{Q}_\phi(y_1|x_0)$ is also a classifier for observations $y_2, \ldots, y_n$.

For a non-terminal state $x_t$ with $t \in (0, T]$, we must train $\widetilde{Q}$ to predict $y_1, \ldots, y_n$ jointly. For a non-terminal state $\widehat{x}_t$ and observations $\widehat{y}_1, \ldots, \widehat{y}_n$, the cross-entropy loss is

$$\ell(\widehat{x}_t, \widehat{y}_1, \ldots, \widehat{y}_n; \phi) = -\log \widetilde{Q}_\phi(y_1 = \widehat{y}_1, \ldots, y_n = \widehat{y}_n|x_t = \widehat{x}_t). \quad (27)$$

Tuples $(\widehat{x}_t, \widehat{y}_1, \ldots, \widehat{y}_n)$ are obtained as follows: 1) $\widehat{y}_1 \sim \widetilde{p}(y_1)$; 2) A trajectory $\tau = \{x_t\}_{t=0}^{T}$ is sampled from the reverse SDE of diffusion model $y_1$. At this point, we would ideally sample $\widehat{y}_2, \ldots, \widehat{y}_n$ given $\widehat{x}_0$ but this requires access to $\widetilde{p}(y_k = \widehat{y}_k|\widehat{x}_0)$. Instead, we approximate this with $w_i(\widehat{x}; \phi) = \widetilde{Q}_\phi(y_1 = i|x_0 = \widehat{x}_0)$ and marginalize over $\widehat{y}_2, \ldots, \widehat{y}_n$ to obtain the cross-entropy loss

$$\mathcal{L}_N(\phi, \overline{\phi}) = \mathbb{E}_{(\widehat{\tau}, \widehat{y}_1) \sim \widetilde{p}(\tau, y_1)} \left[ \sum_{\widehat{x}_t \in \widehat{\tau} \backslash \{\widehat{x}_0\}} \sum_{\widehat{y}_2=1}^{m} \cdots \sum_{\widehat{y}_n=1}^{m} \left( \prod_{k=2}^{n} w_{\widehat{y}_k}(\widehat{x}_0; \overline{\phi}) \right) \ell(\widehat{x}_t, \widehat{y}_1, \ldots, \widehat{y}_n; \phi) \right]. \quad (28)$$

# B Classifier Guidance for Parameterized Operations

This section covers the details of classifier guidance and classifier training for the parameterized operations (Section 4.1).

The complete probabilistic model for the parameterized operations on two distributions is given by

$$\widetilde{p}(x, y_1, y_2; \alpha) = \widetilde{p}(x)\widetilde{p}(y_1|x)\widetilde{p}(y_2|x; \alpha), \quad \widetilde{p}(x) = \frac{1}{2}p_1(x) + \frac{1}{2}p_2(x), \tag{29a}$$

$$\widetilde{p}(y_1 = i|x) = \frac{p_i(x)}{p_1(x) + p_2(x)}, \quad \widetilde{p}(y_2 = i|x; \alpha) = \frac{\left(\alpha p_1(x)\right)^{[i=1]} \cdot \left((1-\alpha)p_2(x)\right)^{[i=2]}}{\alpha p_1(x) + (1 - \alpha)p_2(x)}. \tag{29b}$$

While in the probabilistic model (3) all observations $y_i$ are exchangeable, in the parameterized model (29) $y_1$ and $y_2$ are not symmetric. This difference requires changes in the classifier training algorithm for the parameterized operations.

We develop the method for the parameterized operations based on two observations:

- $y_1$ appears in (29) in the same way as in (3);
- the likelihood $\widetilde{p}(y_2|x; \alpha)$ of $y_2$ given $x$ can be expressed as the function of $\widetilde{p}(y_1|x)$ and $\alpha$:

$$\widetilde{p}(y_2 = 1|x; \alpha) = \frac{\alpha p_1(x)}{\alpha p_1(x) + (1 - \alpha)p_2(x)} = \frac{\alpha \frac{p_1(x)}{p_1(x)+p_2(x)}}{\alpha \frac{p_1(x)}{p_1(x)+p_2(x)} + (1 - \alpha)\frac{p_2(x)}{p_1(x)+p_2(x)}} \tag{30a}$$

$$= \frac{\alpha \widetilde{p}(y_1 = 1|x)}{\alpha \widetilde{p}(y_1 = 1|x) + (1 - \alpha)\widetilde{p}(y_1 = 2|x)},$$

$$\widetilde{p}(y_2 = 2|x; \alpha) = \frac{(1 - \alpha)p_2(x)}{\alpha p_1(x) + (1 - \alpha)p_2(x)} = \frac{(1 - \alpha)\frac{p_2(x)}{p_1(x)+p_2(x)}}{\alpha \frac{p_1(x)}{p_1(x)+p_2(x)} + (1 - \alpha)\frac{p_2(x)}{p_1(x)+p_2(x)}} \tag{30b}$$

$$= \frac{(1 - \alpha)\widetilde{p}(y_1 = 2|x)}{\alpha \widetilde{p}(y_1 = 1|x) + (1 - \alpha)\widetilde{p}(y_1 = 2|x)}.$$

These two observations combined suggest the training procedure where 1) the terminal state classifier is trained to approximate $\widetilde{p}(y_1 = i|x)$ in the same way as in Section 5.2; 2) the probability estimates $w_i(\hat{x}, \hat{\alpha}; \phi) \approx \widetilde{p}(y_2 = i; \hat{\alpha})$ are expressed through the learned terminal state classifier $\widetilde{p}(y_1 = i|x)$ via (30). Below we provide details of this procedure for the case of GFlowNet composition.

**Learning the terminal state classifier.** The marginal $y_1$ classifier $\widetilde{Q}_\phi(y_1|x)$ is learned by minimizing the cross-entropy loss

$$\mathcal{L}_T(\phi) = \mathop{\mathbb{E}}_{(\hat{x},\hat{y}_1) \sim \widetilde{p}(x,y_1)} \left[ -\log \widetilde{Q}_\phi(y_1 = \hat{y}_1|x = \hat{x}) \right]. \tag{31}$$

Then, the joint classifier $\widetilde{Q}_\phi(y_1, y_2|x; \alpha)$ is constructed as

$$\widetilde{Q}_\phi(y_1, y_2|x; \alpha) = \widetilde{Q}_\phi(y_1|x)\widetilde{Q}_\phi(y_2|x; \alpha), \tag{32}$$

where $\widetilde{Q}_\phi(y_2|x; \alpha)$ can be expressed through the marginal $\widetilde{Q}_\phi(y_1|x)$ via (30).

**Learning the non-terminal state classifier.** The non-terminal state classifier $\widetilde{Q}(y_1, y_2|s; \alpha)$ models $y_1$ and $y_2$ jointly. Note that $\alpha$ is one of the inputs to the classifier model. Given a sampled trajectory $\hat{\tau}$, labels $\hat{y}_1, \hat{y}_2$, and $\hat{\alpha}$, the total cross-entropy loss of all non-terminal states in $\hat{\tau}$ is

$$\ell(\hat{\tau}, \hat{y}_1, \hat{y}_2, \hat{\alpha}; \phi) = \sum_{t=0}^{|\tau|-1} \left[ -\log \widetilde{Q}_\phi(y_1 = \hat{y}_1, y_2 = \hat{y}_2|s = \hat{s}_t; \hat{\alpha}) \right]. \tag{33}$$

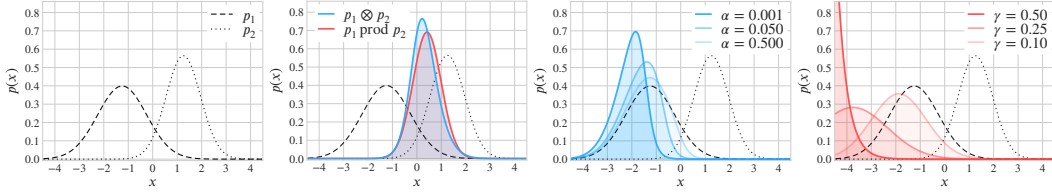

(a) Base distributions $p_1, p_2$    (b) HM$_{\text{(ours)}}$, Product    (c) Contrasts$_{\text{(ours)}}$ $p_1 \, \mathbf{\Phi}_{(1-\alpha)} \, p_2$    (d) Negations $p_1 \, \text{neg}_\gamma \, p_2$

Figure C.1: **Compositional sculpting and energy operations applied to 1D Gaussian distributions.**
(a) Densities of base 1D Gaussian distributions $p_1(x) = \mathcal{N}(x; -5/4, 1)$ and $p_2(x) = \mathcal{N}(x; 5/4, 1/2)$.
(b) harmonic mean $p_1 \otimes p_2$ and product $p_1 \text{ prod } p_2$. (c) parameterized contrasts $p_1 \, \mathbf{\Phi}_{(1-\alpha)} \, p_2$ at different values of $\alpha$. (d) negations $p_1 \, \text{neg}_\gamma \, p_2$ at different values of $\gamma$. Curves show the PDFs of distributions.

The pairs $(\hat{\tau}, \hat{y}_1)$ can be generated via a sampling scheme similar to the one used for the terminal state classifier loss above: 1) $\hat{y}_1 \sim \widetilde{p}(y_1)$ and 2) $\hat{\tau} \sim p_{\hat{y}_1}(\tau)$. An approximation of the distribution of $\hat{y}_2$ given $\hat{\tau}$ is constructed using (30):

$$w_1(\hat{x}, \hat{\alpha}; \phi) = \frac{\hat{\alpha} \widetilde{Q}_\phi(y_1 = 1 | x = \hat{x})}{\hat{\alpha} \widetilde{Q}_\phi(y_1 = 1 | x = \hat{x}) + (1 - \hat{\alpha}) \widetilde{Q}_\phi(y_1 = 2 | x = \hat{x})} \approx \widetilde{p}(y_2 = 1 | x = \hat{x}; \hat{\alpha}), \tag{34a}$$

$$w_2(\hat{x}, \hat{\alpha}; \phi) = \frac{(1 - \hat{\alpha}) \widetilde{Q}_\phi(y_1 = 2 | x = \hat{x})}{\hat{\alpha} \widetilde{Q}_\phi(y_1 = 1 | x = \hat{x}) + (1 - \hat{\alpha}) \widetilde{Q}_\phi(y_1 = 2 | x = \hat{x})} \approx \widetilde{p}(y_2 = 2 | x = \hat{x}; \hat{\alpha}). \tag{34b}$$

Since these expressions involve outputs of the terminal state classifier which is being trained simultaneously, we again (see Section 5.2) introduce the target network parameters $\overline{\phi}$ that are used to compute the probability estimates (34).

The training loss for the non-terminal state classifier is

$$\mathcal{L}_N(\phi, \overline{\phi}) = \underset{\hat{\alpha} \sim p(\alpha)}{\mathbb{E}} \underset{(\hat{\tau}, \hat{y}_1) \sim \widetilde{p}(\tau, y_1)}{\mathbb{E}} \left[ \sum_{\hat{y}_2 = 1}^{2} w_{\hat{y}_2}(\hat{x}, \hat{\alpha}; \overline{\phi}) \ell(\hat{\tau}, \hat{y}_1, \hat{y}_2, \hat{\alpha}; \phi) \right], \tag{35}$$

where $p(\alpha)$ is sampling distribution over $\alpha \in (0, 1)$. In our experiments, we used the following sampling scheme for $\alpha$:

$$\hat{z} \sim U[-B, B], \qquad \hat{\alpha} = \frac{1}{1 + \exp(-\hat{z})}. \tag{36}$$

## C    Analysis of Compositional Sculpting and Energy Operations

The harmonic mean and contrast operations we have introduced are analogous to the product and negation operations for EBMs respectively. Although the harmonic mean and product operations are quite similar in practice, unlike the negation operation our proposed contrast operation always results in a valid probability distribution. Figure C.1 shows the results of these operations applied to two Gaussian distributions. The harmonic mean and product, shown in panel (b), are both concentrated on points that have high probability under both Gaussians. Figure C.1(c) shows parameterized contrasts $p_1 \, \mathbf{\Phi}_{(1-\alpha)} p_2$ at different values of $\alpha$, and panel (d) shows negations $p_1 \, \text{neg}_\gamma \, p_2$ at different values of $\gamma$. The effect of negation at $\gamma = 0.1$ resembles the effect of the contrast operation: the density retreats from the high likelihood region of $p_2$. However, as $\gamma$ increases to 0.5 the distribution starts to concentrate excessively on the values $x < -3$. This is due to the instability of division $p_1(x)/(p_2(x))^\gamma$ in regions where $p_2(x) \to 0$. Proposition C.1 shows that negation $p_1 \, \text{neg}_\gamma \, p_2$ in many cases results in an improper (non-normalizable) distribution.

**Mathematical analysis of operations.** Harmonic mean and product are not defined for pairs of distributions $p_1, p_2$ which have disjoint supports. In such cases, attempts at evaluation of the expressions for $p_1 \otimes p_2$ and $p_1 \text{ prod } p_2$ will lead to impossible probability distributions that have zero

probability mass (density) everywhere[3]. The result of both harmonic mean and product are correctly defined for any pair of distributions $p_1$, $p_2$ that have non-empty support intersection.

Notably, contrast is well-defined for any input distributions while negation is ill-defined for some input distributions $p_1$, $p_2$ as formally stated below (see Figure C.1 (d) for a concrete example).

**Proposition C.1.**

1. *For any $\alpha \in (0, 1)$ the parameterized contrast operation $p_1 \; \unicode{x25D1} \;_{(1-\alpha)} \; p_2$ (7) is well-defined: gives a proper distribution for any pair of distributions $p_1$, $p_2$.*

2. *For any $\gamma \in (0, 1)$ there are infinitely many pairs of distributions $p_1$, $p_2$ such that the negation*

$$(p_1 \; neg_\gamma \; p_2)(x) \propto \exp\left\{-\left(E_1(x) - \gamma E_2(x)\right)\right\} \propto \frac{p_1(x)}{\left(p_2(x)\right)^\gamma}. \tag{37}$$

*results in an improper (non-normalizable) distribution.*

*Proof.* Without loss of generality, we prove the claims of the proposition assuming absolutely continuous distributions $p_1$, $p_2$ with probability density functions $p_1(\cdot)$, $p_2(\cdot)$.

**Claim 1.** For any two distributions $p_1$, $p_2$ we have $p_1(x) \geq 0$, $p_2(x) \geq 0$, $\int_\mathcal{X} p_1(x)\, dx = \int_\mathcal{X} p_2(x)\, dx = 1 < \infty$. Then, the RHS of the expression for the parameterized contrast operation $p_1 \; \unicode{x25D1} \;_{(1-\alpha)} \; p_2$ (7) satisfies

$$\frac{p_1(x)^2}{\alpha p_1(x) + (1-\alpha)p_2(x)} = \frac{p_1(x)}{\alpha} \cdot \underbrace{\frac{p_1(x)}{p_1(x) + \frac{(1-\alpha)}{\alpha}p_2(x)}}_{\leq 1} \leq \frac{p_1(x)}{\alpha}, \quad \forall x \in \text{supp}(p_1) \cup \text{supp}(p_2). \tag{38}$$

For points $x \notin \text{supp}(p_1) \cup \text{supp}(p_2)$, we set $\frac{p_1(x)^2}{\alpha p_1(x)+(1-\alpha)p_2(x)} = 0$ since by construction the composite distributions do not have probability mass outside of the union of the supports of the original distributions. The above implies that

$$\int_\mathcal{X} \frac{p_1^2(x)}{\alpha p_1(x) + (1-\alpha)p_2(x)}\, dx \leq \frac{1}{\alpha} \int_\mathcal{X} p_1(x)\, dx = \frac{1}{\alpha} < \infty. \tag{39}$$

Therefore, the RHS of the expression for the parameterized contrast operation $p_1 \; \unicode{x25D1} \;_{(1-\alpha)} \; p_2$ (7) can be normalized, and the distribution $p_1 \; \unicode{x25D1} \;_{(1-\alpha)} \; p_2$ is well-defined.

**Claim 2.** For any $\gamma \in (0, 1)$ we provide an infinite collection of distribution pairs $p_1$ and $p_2$ such that negation $p_1 \; neg_\gamma \; p_2$ results in a non-normalizable distribution.

For the given $\gamma \in (0, 1)$ we select four numbers $\mu_1 \in \mathbb{R}$, $\mu_2 \in \mathbb{R}$, $\sigma_1 > 0$, $\sigma_2 > 0$ such that

$$\sigma_1^2 \geq \frac{1}{\gamma}\sigma_2^2, \tag{40}$$

Consider univariate normal distributions $p_1(x) = \mathcal{N}(x; \mu_1, \sigma_1^2)$, $p_2(x) = \mathcal{N}(x; \mu_2, \sigma_2^2)$ with density functions

$$p_i(x) = \mathcal{N}(x; \mu_i, \sigma_i^2) = \frac{1}{\sqrt{2\pi\sigma_i^2}} \exp\left\{-\frac{(x - \mu_i)^2}{2\sigma_i^2}\right\}, \qquad i \in \{1, 2\}. \tag{41}$$

For such $p_1$ and $p_2$, the RHS of (37) is

$$\frac{p_1(x)}{(p_2(x))^\gamma} = \frac{1}{\left(\sqrt{2\pi}\right)^{1-\gamma}} \frac{\sigma_2^\gamma}{\sigma_1} \exp\left\{x^2\left(\frac{\gamma\sigma_1^2 - \sigma_2^2}{2\sigma_1^2\sigma_2^2}\right) + x\left(\frac{\mu_1}{\sigma_1^2} - \gamma\frac{\mu_2}{\sigma_2^2}\right) + \gamma\frac{\mu_2^2}{2\sigma_2^2} - \frac{\mu_1^2}{2\sigma_1^2}\right\}. \tag{42}$$

Condition (40) implies that the quadratic function under the exponent above has a non-negative coefficient for $x^2$. Therefore this function either grows unbounded as $x \to \infty$ (if the coefficients for the quadratic and linear terms are not zero), or constant (if the coefficients for quadratic and linear terms are zero). In either case, $\int_\mathbb{R} p_1(x)/(p_2(x))^\gamma \, dx = \infty$. $\qquad\square$

---

[3]Informal interpretation: distributions with disjoint supports have empty "intersections" (think of the intersection of sets analogy)

# D Proofs and Derivations

## D.1 Proof of Proposition 5.1

Our goal is to show that the policy (12) induces the mixture distribution $p_M(x) = \sum_{i=1}^m \omega_i p_i(x)$.

**Preliminaries.** In our proof below we use the notion of "the probability of observing a state $s \in S$ on a GFlowNet trajectory". Following Bengio et al. [35], we abuse the notation and denote this probability by

$$p_i(s) \triangleq p_i(\{\tau : s \in \tau\}) = \sum_{\tau \in \mathcal{T}_{s_0,s}} \prod_{t=0}^{|\tau|-1} p_{i,F}(s_t|s_{t-1}), \tag{43}$$

where $\mathcal{T}_{s_0,s}$ is the set of all (sub)trajectories starting at $s_0$ and ending at $s$. The probabilities induced by the policy (12) are denoted by $p_M(s)$. Note that $p_i(s)$ and $p_M(s)$ should not be interpreted as probability mass functions over the set of states $S$. In particular $p_i(s_0) = p_M(s_0) = 1$ and sums $\sum_{s \in S} p_i(s)$, $\sum_{s \in S} p_M(s)$ are not equal to 1 (unless $S = \{s_0\}$). However, the functions $p_i(\cdot)$, $p_M(\cdot)$ restricted to the set of terminal states $\mathcal{X}$ give valid probability distributions over $\mathcal{X}$: $\sum_{x \in \mathcal{X}} p_i(x) = \sum_{x \in \mathcal{X}} p_M(x) = 1$.

By definition $p_i(\cdot)$ and $p_M(\cdot)$ satisfy the recurrent relationship

$$p_i(s) = \sum_{s_* : (s_* \to s) \in \mathcal{A}} p_i(s_*) p_{i,F}(s|s_*), \qquad p_M(s) = \sum_{s_* : (s_* \to s) \in \mathcal{A}} p_M(s_*) p_{M,F}(s|s_*). \tag{44}$$

The joint distribution of $y$ and $\tau$ described in the statement of Proposition 5.1 is $p(\tau, y{=}i) = w_i p_i(\tau)$. This joint distribution over $y$ and trajectories implies the following expressions for the distributions involving intermediate states $s$.

$$p(y{=}i) = \omega_i, \tag{45}$$

$$p(\tau|y{=}i) = p_i(\tau) = \prod_{t=0}^{|\tau|-1} p_{i,F}(s_t|s_{t-1}), \tag{46}$$

$$p(\tau) = \sum_{i=1}^m p(\tau|y{=}i)p(y{=}i) = \sum_{i=1}^m \omega_i p_i(\tau), \tag{47}$$

$$p(s|y{=}i) = p_i(s), \tag{48}$$

$$p(s) = \sum_{i=1}^m p(s|y{=}i)p(y{=}i) = \sum_{i=1}^m \omega_i p_i(s). \tag{49}$$

**Proof.** Using the notation introduced above, we can formally state our goal. We need to show that $p_M(x)$ induced by $p_{M,F}$ gives the mixture distribution

$$p_M(x) = \sum_{i=1}^m \omega_i p_i(x). \tag{50}$$

We prove a more general equation for all states $s \in S$

$$p_M(s) = \sum_{i=1}^m \omega_i p_i(s) \tag{51}$$

by induction over the DAG $(S, \mathcal{A})$.

**Base case.** Consider the initial state $s_0 \in S$. By definition $p_i(s_0) = p_M(s_0) = 1$ which implies

$$p_M(s_0) = \sum_{i=1}^m \omega_i p_i(s_0). \tag{52}$$

**Inductive step.** Consider a state $s$ such that (51) holds for all predecessor states $s_* : (s_* \to s) \in \mathcal{A}$. For such a state we have

$$p_M(s) = \sum_{s_* : (s_* \to s) \in \mathcal{A}} p_M(s_*) p_{M,F}(s \mid s_*) \quad \{\text{used (44)}\} \tag{53}$$

$$= \sum_{s_* : (s_* \to s) \in \mathcal{A}} p_M(s_*) \left( \sum_{i=1}^{m} p(y = i \mid s_*) p_{i,F}(s \mid s_*) \right) \quad \{\text{used definition of } p_{M,F}\} \tag{54}$$

$$= \sum_{s_* : (s_* \to s) \in \mathcal{A}} \frac{p_M(s_*)}{p(s_*)} \left( \sum_{i=1}^{m} p(s_* \mid y=i) p(y=i) p_{i,F}(s \mid s_*) \right) \quad \{\text{used Bayes' theorem}\} \tag{55}$$

$$= \sum_{s_* : (s_* \to s) \in \mathcal{A}} \frac{p_M(s_*)}{\sum_{i=1}^{m} \omega_i p_i(s_*)} \left( \sum_{i=1}^{m} \omega_i p_i(s_*) p_{i,F}(s \mid s_*) \right) \quad \{\text{used (45), (48), (49)}\} \tag{56}$$

$$= \sum_{s_* : (s_* \to s) \in \mathcal{A}} \left( \sum_{i=1}^{m} \omega_i p_i(s_*) p_{i,F}(s \mid s_*) \right) \quad \{\text{used induction hypothesis}\} \tag{57}$$

$$= \sum_{i=1}^{m} \omega_i \left( \sum_{s_* : (s_* \to s) \in \mathcal{A}} p_i(s_*) p_{i,F}(s \mid s_*) \right) \quad \{\text{changed summation order}\} \tag{58}$$

$$= \sum_{i=1}^{m} \omega_i p_i(s), \quad \{\text{used (44)}\} \tag{59}$$

which proves (51) for $s$.

## D.2   Proof of Proposition 5.2

**Claim 1.** Our goal is to prove the relationship (13) for all non-terminal states $s \in \mathcal{S} \setminus \mathcal{X}$. To prove this relationship, we invoke several important properties of Markovian probability flows on DAGs [35].

By Proposition 16 of Bengio et al. [35] for the given GFlowNet forward policy $p_F(\cdot | \cdot)$ there exists a unique backward policy $p_B(\cdot | \cdot)$ such that the probability of any complete trajectory $\tau = (s_0 \to \ldots \to s_{|\tau|} = x)$ in DAG $(\mathcal{S}, \mathcal{A})$ can be expressed as

$$p(\tau) = p(x) \prod_{t=1}^{|\tau|} p_B(s_{t-1} | s_t), \tag{60}$$

and the probability of observing a state $s \in \mathcal{S}$ on a trajectory can be expressed as

$$p(s) = \sum_{x \in \mathcal{X}} p(x) \sum_{\tau \in \mathcal{T}_{s,x}} \prod_{t=1}^{|\tau|} p_B(s_{t-1} | s_t), \tag{61}$$

where $\mathcal{T}_{s,x}$ is the set of all (sub)trajectories starting at $s$ and ending at $x$. Moreover, $p_F(\cdot | \cdot)$ and $p_B(\cdot | \cdot)$ are related through the "detailed balance condition" [35, Proposition 21]

$$p(s) p_F(s' | s) = p(s') p_B(s | s'), \qquad \forall (s \to s') \in \mathcal{A}. \tag{62}$$

By the statement of Proposition 5.2, in the probabilistic model $p(x, y)$, the marginal distribution $p(x)$ is realized by the GFlowNet forward policy $p_F(\cdot | \cdot)$ and $y$ is independent of intermediate states $s$. The joint distribution $p(s, y)$ is given by

$$p(s, y) = \sum_{x \in \mathcal{X}} p(x, y) \sum_{\tau \in \mathcal{T}_{s,x}} \prod_{t=1}^{|\tau|} p_B(s_{t-1} | s_t) \tag{63}$$

$$= \sum_{x \in \mathcal{X}} p(x, y) \sum_{s' : (s \to s') \in \mathcal{A}} p_B(s | s') \sum_{\tau \in \mathcal{T}_{s',x}} \prod_{t=1}^{|\tau|} p_B(s_{t-1} | s_t) \tag{64}$$

$$= \sum_{s':(s\to s')\in\mathcal{A}} p_B(s|s') \underbrace{\sum_{x\in\mathcal{X}} p(x,y) \sum_{\tau\in\mathcal{T}_{s',x}} \prod_{t=1}^{|\tau|} p_B(s_{t-1}|s_t)}_{p(s',y)} \qquad (65)$$

$$= \sum_{s':(s\to s')\in\mathcal{A}} p_B(s|s')p(s',y). \qquad (66)$$

Expressing the conditional probability $p(y|s)$ through the joint $p(s,y)$ we obtain

$$p(y|s) = \frac{p(s,y)}{p(s)} \quad \{\text{used definition of conditional probability}\} \qquad (67)$$

$$= \frac{1}{p(s)} \sum_{s':(s\to s')\in\mathcal{A}} p_B(s|s')p(s',y) \quad \{\text{used (66)}\} \qquad (68)$$

$$= \sum_{s':(s\to s')\in\mathcal{A}} p_B(s|s')\frac{p(s')}{p(s)}p(y|s') \quad \{\text{decomposed } p(s',y)=p(s')p(y|s')\} \qquad (69)$$

$$= \sum_{s':(s\to s')\in\mathcal{A}} p_F(s'|s)p(y|s'), \quad \{\text{used (62)}\} \qquad (70)$$

which proves (13).

**Claim 2**. Our goal is to show that the classifier-guided policy (14) induces the conditional distribution $p(y|x)$.

We know (Section D.1) that the state probabilities induced by the marginal GFlowNet policy $p_F(\cdot|\cdot)$ satisfy the recurrence

$$p(s) = \sum_{s_*:(s_*\to s)\in\mathcal{A}} p(s_*)p_F(s|s_*). \qquad (71)$$

Let $p_y(\cdot)$ denote the state probabilities induced by the classifier-guided policy (14). These probabilities by definition (Section D.1) satisfy the recurrence

$$p_y(s) = \sum_{s_*:(s_*\to s)\in\mathcal{A}} p_y(s_*)p_F(s|s_*,y). \qquad (72)$$

We show that

$$p_y(s) = p(s|y), \qquad (73)$$

by induction over DAG $(\mathcal{S},\mathcal{A})$.

**Base case.** Consider the initial state $s_0$. By definition $p_y(s_0) = 1$. At the same time $p(s_0|y) = p(\{\tau : s_0 \in \tau\}|y) = 1$. Therefore $p_y(s_0) = p(s_0|y)$.

**Inductive step.** Consider a state $s$ such that (73) holds for all predecessor states $s_* : (s_* \to s) \in \mathcal{A}$. For such a state we have

$$p_y(s) = \sum_{s_*:(s_*\to s)\in\mathcal{A}} p_y(s_*)p_F(s|s_*,y) \quad \{\text{used (72)}\} \qquad (74)$$

$$= \sum_{s_*:(s_*\to s)\in\mathcal{A}} p_y(s_*)p_F(s|s_*)\frac{p(y|s)}{p(y|s_*)} \quad \{\text{used (14)}\} \qquad (75)$$

$$= \sum_{s_*:(s_*\to s)\in\mathcal{A}} p(s_*|y)p_F(s|s_*)\frac{p(y|s)}{p(y|s_*)} \quad \{\text{used induction hypothesis}\} \qquad (76)$$

$$= \sum_{s_*:(s_*\to s)\in\mathcal{A}} \frac{p(y|s_*)p(s_*)}{p(y)}p_F(s|s_*)\frac{p(y|s)}{p(y|s_*)} \quad \{\text{used Bayes' theorem}\} \qquad (77)$$

$$= \frac{p(y|s)}{p(y)} \sum_{s_*:(s_*\to s)\in\mathcal{A}} p(s_*)p_F(s|s_*) \quad \{\text{rearranged terms}\} \qquad (78)$$

$$= \frac{p(y|s)}{p(y)} p(s) \quad \{\text{used (71)}\} \tag{79}$$

$$= p(s|y), \quad \{\text{used Bayes' theorem}\} \tag{80}$$

which proves (73) for state $s$.

### D.3 Proof of Theorem 5.3

By Proposition 5.1 we have that the policy

$$p_{M,F}(s'|s) = \sum_{i=1}^{m} p_{i,F}(s'|s)\widetilde{p}(y=i|s), \tag{81}$$

generates the mixture distribution $p_M(x) = \frac{1}{m} \sum_{i=1}^{m} p_i(x)$.

In the probabilistic model $\widetilde{p}(x, y_1 \dots, y_n)$ the marginal distribution $\widetilde{p}(x) = p_M(x)$ is realized by the mixture policy $p_{M,F}$. Therefore, $\widetilde{p}(x, y_1, \dots, y_n)$ satisfies the conditions of Proposition 5.2 which states that the conditional distribution $\widetilde{p}(x|y_1, \dots, y_n)$ is realized by the classifier-guided policy

$$p_F(s'|s, y_1, \dots, y_n) = p_{M,F}(s'|s)\frac{\widetilde{p}(y_1, \dots, y_n|s')}{\widetilde{p}(y_1, \dots, y_n|s)} = \frac{\widetilde{p}(y_1, \dots y_n \mid s')}{\widetilde{p}(y_1, \dots y_n \mid s)} \sum_{i=1}^{m} p_{i,F}(s'|s)\widetilde{p}(y=i|s). \tag{82}$$

### D.4 Detailed Derivation of Classifier Training Objective

This section provides a more detailed step-by-step derivation of the non-terminal state classifier training objective (18).

**Step 1.** Our goal is to train a classifier $\widetilde{Q}(y_1, \dots, y_n|s)$. This classifier can be obtained as the optimal solution of

$$\min_{\phi} \mathbb{E}_{\widehat{\tau}, \widehat{y}_1, \dots, \widehat{y}_n \sim \widetilde{p}(\tau, y_1, \dots, y_n)}\left[\ell(\widehat{\tau}, \widehat{y}_1, \dots, \widehat{y}_n; \phi)\right], \tag{83}$$

where $\ell(\cdot)$ is defined in equation (17). An unbiased estimate of the loss (and its gradient) can be obtained by sampling $(\widehat{\tau}, \widehat{y}_1, \dots, \widehat{y}_n)$ and evaluating (17) directly. However sampling tuples $(\tau, y_1, \dots, y_n)$ is not straightforward. The following steps describe our proposed approach to the estimation of expectation in (83).

**Step 2.** The expectation in (83) can be expressed as

$$\mathbb{E}_{\widehat{\tau}, \widehat{y}_1 \sim \widetilde{p}(\tau, y_1)}\left[\sum_{\widehat{y}_2=1}^{m} \cdots \sum_{\widehat{y}_n=1}^{m}\left(\prod_{i=2}^{n}\widetilde{p}(y_i=\widehat{y}_i|x=\widehat{x})\right)\ell(\widehat{\tau}, \widehat{y}_1, \dots, \widehat{y}_n; \phi)\right], \tag{84}$$

where we re-wrote the expectation over $(y_2, \dots, y_n)|\tau$ as the explicit sum of the form $\mathbb{E}_{q(z)}[g(z)] = \sum_{z \in \mathcal{Z}} q(z)g(z)$. The expectation over $(\tau, y_1)$ can be estimated by sampling pairs $(\widehat{\tau}, \widehat{y}_1)$ as described in the paragraph after equation (17): 1) $\widehat{y}_1 \sim \widetilde{p}(y_1)$ and 2) $\widehat{\tau} \sim p_{\widehat{y}_1}(\tau)$. The only missing part is the probabilities $\widetilde{p}(y_i=\widehat{y}_i|x=\widehat{x})$ which are not directly available.

**Step 3.** Our proposal is to approximate these probabilities as $\widetilde{p}(y_1=j|x=\widehat{x}) \approx w_j(\widehat{x}; \phi) = \widetilde{Q}_\phi(y_1=j|x=\widehat{x})$. The idea here is that the terminal state classifier $\widetilde{Q}_\phi(y_1|x)$, when trained to optimality, produces outputs exactly equal to the probabilities $\widetilde{p}(y_1|x)$, and the more the classifier is trained the better is the approximation of the probabilities.

**Step 4.** Steps 1-3, give a procedure where the computation of the non-terminal state classification loss requires access to the terminal state classifier. As we described in the paragraph preceding equation (18), we propose to train non-terminal and terminal classifiers simultaneously and introduce "target network" parameters. The weights $w$ are computed by the target network $\widetilde{Q}_{\bar{\phi}}$.

Combining all the steps above, we arrive at objective (18) which we use to estimate the expectation in (83).

Note that equation (18) involves summation over $\widehat{y}_2, \ldots \widehat{y}_n$ with $m^{n-1}$ terms in the sum. If values of $n$ and $m$ are small, the sum can be evaluated directly. In general, one could trade off estimation accuracy for improved speed by replacing the summation with Monte Carlo estimation. In this case, the values $\widehat{y}_k$ are sampled from the categorical distributions $Q_{\overline{\phi}}(y|x)$. Note that labels can be sampled in parallel since $y_i$ are independent given $x$.

## D.5 Assumptions and Proof of Proposition A.1

This subsection provides a formal statement of the assumptions and a more detailed formulation of Proposition A.1.

The assumptions, the formulation of the result, and the proof below closely follow those of Theorem 1 of Peluchetti [53]. Theorem 1 in [53] generalizes the result of Brigo [57] (Corollary 1.3), which derives the SDE for mixtures of 1D diffusion processes.

We found an error in the statement and the proof of Theorem 1 (Appendix A.2 of [53]). The error makes the result of [53] for $D$-dimensional diffusion processes disagree with the result of [57] for 1-dimensional diffusion processes.

Here we provide a corrected version of Theorem 1 of [53] in a modified notation and a simplified setting (mixture of finite rather than infinite number of diffusion processes). Most of the content is directly adapted from [53].

Notation:

- for a vector-valued $f : \mathbb{R}^D \to \mathbb{R}^D$, the divergence of $f$ is denoted as $\nabla \cdot (f(x)) = \sum_{d=1}^{D} \frac{\partial}{\partial x_d} f_d(x)$,

- for a scalar-values $a : \mathbb{R}^D \to \mathbb{R}$, the divergence of the gradient of $a$ (the Laplace operator) is denoted by $\Delta(a(x)) = \nabla \cdot (\nabla a(x)) = \sum_{d=1}^{D} \frac{\partial^2}{\partial x_d^2} a(x)$.

**Assumption 1** (SDE solution). A given $D$-dimensional SDE$(f, g)$:

$$dx_t = f_t(x_t)dt + g_t dw_t, \tag{85}$$

with associated initial distribution $p_0(x)$ and integration interval $[0, T]$ admits a unique strong solution on $[0, T]$.

**Assumption 2** (SDE density). A given $D$-dimensional SDE$(f, g)$ with associated initial distribution $p_0(x)$ and integration interval $[0, T]$ admits a marginal density on $(0, T)$ with respect to the $D$-dimensional Lebesgue measure that uniquely satisfies the Fokker-Plank (Kolmogorov-forward) partial differential equation (PDE):

$$\frac{\partial p_t(x)}{\partial t} = -\nabla \cdot (f_t(x)p_t(x)) + \frac{1}{2}\Delta(g_t^2 p_t(x)). \tag{86}$$

**Assumption 3** (positivity). For a given stochastic process, all finite-dimensional densities, conditional or not, are strictly positive.

**Theorem D.1** (Diffusion mixture representation). *Consider the family of D-dimensional SDEs on $t \in [0, T]$ indexed by $i \in \{1, \ldots, m\}$,*

$$dx_{i,t} = f_{i,t}(x_{i,t})dt + g_{i,t}dw_{i,t}, \qquad x_{i,0} \sim p_{i,0}, \tag{87}$$

*where the initial distributions $p_{i,0}$ and the Wiener processes $w_{i,t}$ are all independent. Let $p_{i,t}$, $t \in (0, T)$ denote the marginal density of $x_{i,t}$. For mixing weights $\{\omega_i\}_{i=1}^{m}$, $\omega_i \geq 0$, $\sum_{i=1}^{m} \omega_i = 1$, define the mixture marginal density $p_{M,t}$ for $t \in (0, T)$ and the mixture initial distribution $p_{M,0}$ by*

$$p_{M,t}(x) = \sum_{i=1}^{m} \omega_i p_{i,t}(x) \quad p_{M,0}(x) = \sum_{i=1}^{m} \omega_i p_{M,0}(x). \tag{88}$$

*Consider the D-dimensional SDE on $t \in [0, T]$ defined by*

$$f_{M,t}(x) = \frac{\sum_{i=1}^{m} \omega_i p_{i,t}(x) f_{i,t}(x)}{p_{M,t}(x)}, \qquad g_{M,t}(x) = \sqrt{\frac{\sum_{i=1}^{m} \omega_i p_{i,t}(x) g_{i,t}^2}{p_{M,t}(x)}}, \tag{89}$$

$$dx_t = f_{M,t}(x_t)dt + g_{M,t}(x_t)dw_t, \qquad x_{M,0} \sim p_{M,0}. \tag{90}$$

*It is assumed that all diffusion processes $x_{i,t}$ and the diffusion process $x_{M,t}$ satisfy the regularity Assumptions 1, 2, and 3. Then the marginal distribution of the diffusion $x_{M,t}$ is $p_{M,t}$.*

*Proof.* For $0 < t < T$ we have that

$$\frac{\partial p_{M,t}(x)}{\partial t} = \frac{\partial}{\partial t}\left( \sum_{i=1}^{m} \omega_i p_{i,t}(x) \right) \tag{91}$$

$$= \sum_{i=1}^{m} \omega_i \frac{\partial p_{i,t}(x)}{\partial t} \tag{92}$$

$$= \sum_{i=1}^{m} \omega_i \left( -\nabla \cdot (f_{i,t}(x)p_{i,t}(x)) + \frac{1}{2}\Delta(g_{i,t}^2 p_{i,t}(x)) \right) \tag{93}$$

$$= \sum_{i=1}^{m} \omega_i \left( -\nabla \cdot \left( \frac{p_{i,t}(x)f_{i,t}(x)}{p_{M,t}(x)} p_{M,t}(x) \right) + \frac{1}{2}\Delta\left( \frac{p_{i,t}(x)g_{i,t}^2}{p_{M,t}(x)} p_{M,t}(x) \right) \right) \tag{94}$$

$$= -\nabla \cdot \left( \sum_{i=1}^{m} \frac{\omega_i p_{i,t}(x)f_{i,t}(x)}{p_{M,t}(x)} p_{M,t}(x) \right) + \frac{1}{2}\Delta\left( \sum_{i=1}^{m} \frac{\omega_i p_{i,t}(x)g_{i,t}^2}{p_{M,t}(x)} p_{M,t}(x) \right) \tag{95}$$

$$= -\nabla \cdot (f_{M,t}(x)p_{M,t}(x)) + \frac{1}{2}\Delta(g_{M,t}^2 p_{M,t}(x)). \tag{96}$$

The second is an exchange of the order of summation and differentiation, the third line is the application of the Fokker-Planck PDEs for processes $x_{i,t}$, the fourth line is a rewriting in terms of $p_{M,t}$, the fifth line is another exchange of the order of summation and differentiation. The result follows by noticing that $p_{M,t}(x)$ satisfies the Fokker-Planck equation of $\text{SDE}(f_M, g_M)$. $\qquad\square$

**Proof of Proposition A.1.** Below, we show that the result of Proposition A.1 follows from Theorem D.1.

First, we rewrite $f_{M,t}(x_t)$ and $g_{M,t}(x_t)$ in (89) in terms of the classifier probabilities (21):

$$f_{M,t}(x_t) = \frac{\sum_{i=1}^{m} \omega_i p_{i,t}(x_t)f_{i,t}(x_t)}{p_{M,t}(x_t)} = \sum_{i=1}^{m} p(y=i|x_t)f_{i,t}(x_t), \tag{97}$$

$$g_{M,t}(x_t) = \sqrt{\frac{\sum_{i=1}^{m} \omega_i p_{i,t}(x_t)g_{i,t}^2}{p_{M,t}(x_t)}} = \sqrt{\sum_{i=1}^{m} p(y=i|x_t)g_{i,t}^2}. \tag{98}$$

With these expressions, we apply the result of Theorem D.1 to the base forward processes $dx_{i,t} = f_{i,t}(x_{i,t})\,dt + g_{i,t}\,dw_{i,t}$ and obtain the mixture forward process in equation (19).

From the forward process, we derive the backward process following Song et al. [34]. Using the result of Anderson [38], the backward process for (19) is given by

$$dx_t = \left[ f_{M,t}(x_t) - \nabla_{x_t}(g_{M,t}^2(x_t)) - g_{M,t}^2(x_t)\nabla_{x_t}\log p_{M,t}(x_t) \right] dt + g_{M,t}(x_t)\,d\overline{w}_t. \tag{99}$$

Note that the term $\nabla_{x_t}(g_{M,t}^2(x_t))$ is due to the fact that the diffusion coefficient $g_{M,t}(x_t)$ in (19) is a function of $x$ (cf., equation (16), Appendix A in [34]). This term can be transformed as follows

$$\nabla_{x_t}(g_{M,t}^2(x_t)) = \nabla_{x_t}\left( \sum_{i=1}^{m} p(y=i|x_t)g_{i,t}^2 \right) \tag{100}$$

$$= \sum_{i=1}^{m} g_{i,t}^2 \nabla_{x_t} p(y=i|x_t) \tag{101}$$

$$= \sum_{i=1}^{m} p(y=i|x_t)g_{i,t}^2 \nabla_{x_t}\log p(y=i|x_t) \tag{102}$$

$$= \sum_{i=1}^{m} p(y{=}i|x_t) g_{i,t}^2 \nabla_{x_t} \Big( \log \omega_i + \log p_{i,t}(x_t) - \log p_{M,t}(x_t) \Big) \tag{103}$$

$$= \sum_{i=1}^{m} p(y{=}i|x_t) g_{i,t}^2 \Big( \nabla_{x_t} \log p_{i,t}(x_t) - \nabla_{x_t} \log p_{M,t}(x_t) \Big) \tag{104}$$

$$= \left( \sum_{i=1}^{m} p(y{=}i|x_t) g_{i,t}^2 s_{i,t}(x_t) \right) - g_{M,t}^2(x_t) \nabla_{x_t} \log p_{M,t}(x_t). \tag{105}$$

Substituting the last expression in (99), we notice that the term $g_{M,t}^2(x_t) \nabla_{x_t} \log p_{M,t}(x_t)$ cancels out, and, after simple algebraic manipulations, we arrive at (20).

### D.6 Proof of Theorem A.2

We proove Theorem A.2 in the assumptions of Section D.5. In Section D.5 we established that the mixture diffusion process has the forward SDE

$$dx_t = f_{M,t}(x_t)dt + g_{M,t}(x_t)\, dw_t. \tag{106}$$

and the backward SDE

$$dx_t = \left[ f_{M,t}(x_t) - \nabla_{x_t}(g_{M,t}^2(x_t)) - g_{M,t}^2(x_t) \nabla_{x_t} \log p_{M,t}(x_t) \right] dt + g_{M,t}(x_t)\, d\overline{w}_t. \tag{107}$$

We apply classifier guidance with classifier $\widetilde{p}(y_1, \dots, y_n|x_t)$ to the mixture diffusion process, following Song et al. [34] (see equations (48)-(49) in [34]). The backward SDE of the classifier-guided mixture diffusion is

$$dx_t = \left[ f_{M,t}(x_t) - \nabla_{x_t}(g_{M,t}^2(x_t)) - g_{M,t}^2(x_t) \Big( \nabla_{x_t} \log p_{M,t}(x_t) + \nabla_{x_t} \log \widetilde{p}(y_1, \dots, y_n|x_t) \Big) \right] dt \tag{108}$$

$$+ g_{M,t}(x_t)\, d\overline{w}_t. \tag{109}$$

Finally, we arrive at (23) by substituting (105) in the above, canceling out the term $g_{M,t}^2(x_t) \nabla_{x_t} \log p_{M,t}(x_t)$, and applying simple algebraic manipulations.

## E   Implementation Details

### E.1   Classifier Guidance in GFlowNets

Classifier guidance in GFlowNets (14) is realized through modification of the base forward policy via the multiplication by the ratio of the classifier outputs $p(y|s')/p(y|s)$. The ground truth (theoretically optimal) non-terminal state classifier $p(y|s)$ by Proposition 5.2 satisfies (13) which ensures that the guided policy (14) is valid, i.e. for any state $s \in \mathcal{S}$

$$\sum_{s':(s\to s')\in\mathcal{A}} p_F(s'|s, y) = \sum_{s':(s\to s')\in\mathcal{A}} p_F(s'|s) \frac{p(y|s')}{p(y|s)} \tag{110}$$

$$= \frac{1}{p(y|s)} \underbrace{\sum_{s':(s\to s')\in\mathcal{A}} p_F(s'|s) p(y|s')}_{=p(y|s) \text{ by Proposition 5.2}} = 1. \tag{111}$$

In practice, the ground truth values of $p(y|s)$ are unavailable. Instead, an approximation $Q_\phi(y|s) \approx p(y|s)$ is learned. Equation (13) might not hold for the learned classifier $Q_\phi$, but we still wish to use $\widetilde{Q}_\phi$ for classifier guidance in practice. In order to ensure that the classifier-guided policy is valid in practice even when the approximation $Q_\phi(y|s)$ of the classifier $p(y|s)$ is used, we implement guidance as described below.

First, we express the guided policy (14) in terms of log-probabilities:

$$\log p_F(s'|s, y) = \log p_F(s'|s) + \log p(y|s') - \log p(y|s). \tag{112}$$

Parameterizing distributions through log-probabilities is common practice in probabilistic modeling: GFlowNet forward policies [36, 58] and probabilistic classifiers are typically parameterized by deep neural networks that output logits (unnormalized log-probabilities).

Second, in the log-probability parameterization the guided policy (14) can be equivalently expressed as

$$p_F(s'|s, y) = \left[ \text{softmax} \left( \log p_F(\cdot|s) + \log p(y|\cdot) - \log p(y|s) \right) \right]_{s'} \tag{113}$$

$$= \frac{\exp \left( \log p_F(s'|s) + \log p(y|s') - \log p(y|s) \right)}{\sum_{s'':(s \to s'') \in \mathcal{A}} \exp \left( \log p_F(s''|s) + \log p(y|s'') - \log p(y|s) \right)}. \tag{114}$$

In theory, the softmax operation can be replaced with simple exponentiation, i.e. the numerator in (114) is sufficient on its own since Proposition 5.2 ensures that the sum in the denominator equals to 1. However, using the softmax is beneficial in practice when we substitute learned classifier $Q_\phi(y|s)$ instead of the ground truth classifier $p(y|s)$. Indeed when $Q_\phi(y|s)$ does not satisfy (13), the softmax operation ensures that the guided policy

$$p_F(s'|s, y) = \left[ \text{softmax} \left( \log p_F(\cdot|s) + \log Q_\phi(y|\cdot) - \log Q_\phi(y|s) \right) \right]_{s'} \tag{115}$$

is valid (i.e. probabilities sum up to 1 over $s'$). The fact the softmax expression is valid in theory ensures that policy (115) guided by $Q_\phi(y|s)$ approaches the ground truth policy (guided by $p(y|s)$) as $Q_\phi(y|s)$ approaches $p(y|s)$ throughout training.

## F  Experiment details

### F.1  2D Distributions with GFlowNets

The base GFlowNet forward policies $p_{i,F}(s'|s; \theta)$ were parameterized as MLPs with 2 hidden layers and 256 units in each hidden layer. The cell coordinates of a state $s$ on the 2D $32 \times 32$ grid were one-hot encoded. The dimensionality of the input was $2 \cdot 32$. The outputs of the forward policy network were the logits of the softmax distribution over 3 action choices: 1) move down; 2) move right; 3) stop.

We trained base GFlowNets with the trajectory balance loss [58]. We fixed the uniform backward policy in the trajectory balance objective. We used Adam optimizer [59] with learning rate 0.001, and pre-train the base models for 20 000 steps with batch size 16 (16 trajectories per batch). The log of the total flow $\log Z_\theta$ was optimized with Adam with a learning rate 0.1. In order to promote exploration in trajectory sampling for the trajectory balance objective, we used the sampling policy which takes random actions (uniformly) with probability 0.05 but, otherwise, follows the current learned forward policy.

The classifier was parameterized as MLP with 2 hidden layers and 256 units in each hidden layer. The inputs to the classifier were one-hot encoded cell coordinates, terminal state flag ($\{0, 1\}$), and $\log(\alpha/(1-\alpha))$ (in the case of parameterized operations). The classifier outputs were the logits of the joint label distribution $\widetilde{Q}_\phi(y_1, \ldots, y_n|s)$ for non-terminal states $s$ and the logits of the marginal label distribution $\widetilde{Q}_\phi(y_1|x)$ for terminal states $x$.

We trained the classifier with the loss described in Section 5.2. We used Adam with learning rate 0.001. We performed 15 000 training steps with batch size 64 (64 trajectories sampled from each of the base models per training step). We updated the target network parameters $\overline{\phi}$ as the exponential moving average (EMA) of $\phi$ with the smoothing factor 0.995. We linearly increased the weight $\gamma(\text{step})$ of the non-terminal state loss from 0 to 1 throughout the first 3 000 steps and kept constant $\gamma = 1$ afterward. For the $\alpha$-parameterized version of the classifier (Section B), we used the following sampling scheme for $\alpha$: $\hat{z} \sim U[-3.5, 3.5]$, $\hat{\alpha} = \frac{1}{1+\exp(-\hat{z})}$.

**Quantitative evaluation.**  For each of the composite distributions shown in Figure 2 we evaluated the L1-distance $L_1(p_{\text{method}}, p_{\text{GT}}) = \sum_{x \in \mathcal{X}} |p_{\text{method}}(x) - p_{\text{GT}}(x)|$ between the distribution $p_{\text{method}}$ induced by the classifier-guided policy and the ground-truth composition distribution $p_{\text{GT}}$ computed from the known base model probabilities $p_i$. The evaluation results are presented below.

Figure 2 **top row**. $p_1 \otimes p_2$: $L_1 = 0.071$; $p_1 \,\text{◑}\, p_2$: $L_1 = 0.086$; $p_1 \,\text{◑}_{0.95}\, p_2$: $L_1 = 0.167$.

Figure 2 **bottom row**. $\widetilde{p}(x|y_1 = 1, y_2 = 2)$: $L_1 = 0.076$; $\widetilde{p}(x|y_1 = 1, y_2 = 2, y_3 = 3)$: $L_1 = 0.087$; $\widetilde{p}(x|y_1 = 2, y_2 = 2)$: $L_1 = 0.112$; $\widetilde{p}(x|y_1 = 2, y_2 = 2, y_3 = 2)$: $L_1 = 0.122$.

Figure G.6 shows the distance between the composition and the ground truth as a function of the number of training steps for the classifier as well as the terminal and non-terminal classifier learning curves.

## F.2  Molecule Generation

**Domain.**    In the molecule generation task [36], the objects $x \in \mathcal{X}$ are molecular graphs. The non-terminal states $s \in \mathcal{S} \setminus \mathcal{X}$ are incomplete molecular graphs. The transitions from a given non-terminal state $s$ are of two types: 1) fragment addition $s \rightarrow s'$: new molecular graph $s'$ is obtained by attaching a new fragment to the molecular graph $s$; 2) stop action $s \rightarrow x$: if $s \neq s_0$, then the generation process can be terminated at the molecular graph corresponding to the current state (note that new terminal state $x \in \mathcal{X}$ is different from $s \in \mathcal{S} \setminus \mathcal{X}$, but both states correspond to the same molecular graph).

**Rewards.**    We trained GFlowNets using 3 reward functions: **SEH**, a reward computed by an MPNN [60] that was trained by Bengio et al. [36] to estimate the binding energy of a molecule to the soluble epoxide hydrolase protein; **SA**, an estimate of synthetic accessibility [61] computed with tools from RDKit library [62]; **QED**, a quantitative estimate of drug-likeness [63] which is also computed with RDKit. We normalized all reward functions to the range $[0, 1]$. Higher values of SEH, SA, and QED correspond to stronger binding, higher synthetic accessibility, and higher drug-likeness respectively. Following Bengio et al. [36], we introduced the parameter $\beta$ which controls the sharpness (temperature) of the target distribution: $p(x) \propto R(x)^\beta$, increasing $\beta$ results in a distribution skewed towards high-reward objects. We experimented with two $\beta$ values, 32 and 96 (Figure 3(a),3(d)).

**Reward normalization.**    We used the following normalization rules for SEH, SA, and QED rewards in the molecule domain.

- SEH $= {}^{\text{SEH}_{\text{raw}}}/_{8}$;
- SA $= \frac{10 - \text{SA}_{\text{raw}}}{9}$;
- QED $= \text{QED}_{\text{raw}}$.

**Training and evaluation.**    After training the base GFlowNets with the reward functions described above, we trained classifiers with Algorithm A.1. The classifier was parameterized as a graph neural network based on a graph transformer architecture [64]. Compared to the 2D grid domain (Section 6, "2D distributions via GFlowNet"), we can not directly evaluate the distributions obtained by our approach. Instead, we analyzed the samples generated by the composed distributions. We sampled 5 000 molecules from each composed distribution obtained with our approach as well as the base GFlowNets. We evaluated the sample collections with the two following strategies. **Reward evaluation** (Figure 3, Table 1) we analyzed the distributions of rewards across the sample collections. The goal is to see whether the composition of GFlowNets trained for different rewards leads to noticeable changes in reward distribution. **Distribution distance evaluation** (Figure 4, Table G.5) : we used the samples to estimate the pairwise distances between the distributions. Specifically, for a given pair of distributions represented by two collections of samples $\mathcal{D}_A = \{x_{A,i}\}_{i=1}^n$, $\mathcal{D}_B = \{x_{B,i}\}_{i=1}^n$ we computed the earth mover's distance $d(\mathcal{D}_A, \mathcal{D}_B)$ with ground molecule distance given by $d(x, x') = (\max\{s(x, x'), 10^{-3}\})^{-1} - 1$, where $s(x, x') \in [0, 1]$ is the Tanimoto similarity over Morgan fingerprints of molecules $x$ and $x'$.

**Training details and hyperparameters.**    The base GFlowNet policies were parameterized as graph neural networks with Graph Transformer architecture [64]. We used 6 transformer layers with an embedding of size 128. The input to the Graph Transformer was the graph of fragments with node attributes describing fragments and edge attributes describing attachment points of the edges.

The base GFlowNets were trained with trajectory balance loss. We used Adam optimizer. For the policy network $p_F(s'|s; \theta)$, we set the initial learning rate 0.0005 and exponentially decayed with the factor $2^{\text{step}/20\,000}$. For the log of the total flow $\log Z_\theta$ we set the initial learning rate 0.0005 and

exponentially decayed with the factor $2^{\text{step}/50\,000}$. We trained the base GFlowNets for 15 000 steps with batch size 64 (64 trajectories per batch). In order to promote exploration in trajectory sampling for the trajectory balance objective, we used the sampling policy which takes random actions (uniformly) with probability 0.1 but, otherwise, follows the current learned forward policy.

The classifier was parameterized as a graph neural network with Graph Transformer architecture. We used 4 transformer layers with embedding size 128. The inputs to the classifier were the fragment graph, terminal state flag ($\{0, 1\}$), and $\log(\alpha/(1-\alpha))$ (in case of parameterized operations). The classifier outputs were the logits of the joint label distribution $\widetilde{Q}_\phi(y_1, \ldots, y_n|s)$ for non-terminal states $s$ and the logits of the marginal label distribution $\widetilde{Q}_\phi(y_1|x)$ for terminal states $x$.

We trained the classifier with the loss described in Section 5.2. We used Adam with learning rate 0.001. We performed 15 000 training steps with batch size 8 (8 trajectories sampled from each of the base models per training step). We updated the target network parameters $\overline{\phi}$ as the exponential moving average (EMA) of $\phi$ with the smoothing factor 0.995. We linearly increased the weight $\gamma$(step) of the non-terminal state loss from 0 to 1 throughout the first 4 000 steps and kept constant $\gamma = 1$ afterward. For the $\alpha$-parameterized version of the classifier (Section B), we used the following sampling scheme for $\alpha$: $\hat{z} \sim U[-5.5, 5.5]$, $\hat{\alpha} = \frac{1}{1+\exp(-\hat{z})}$.

## F.3 Colored MNIST Generation via Diffusion Models

The colored MNIST experiment in Section 6 ("Colored MNIST generation via diffusion models") follows the method for composing diffusion models introduced in Appendix A.2. The three base diffusion models were trained on colored MNIST digits generated from the original MNIST dataset. These colored digits were created by mapping MNIST images from their grayscale representation to either the red or green channel, leaving the other channels set to 0. For Figure 5 we post-processed the red and green images generated by the base models and compositions into beige and cyan respectively, which are more accessible colors for colorblind people.

**Models, training details, and hyperparameters.** The base diffusion models were defined as VE SDEs [34]. Their score models were U-Net [65] networks consisting of 4 convolutional layers with 64, 128, 256, and 256 channels and 4 matching transposed convolutional layers. Time was encoded using 256-dimensional Gaussian random features [66]. The score model was trained using Adam optimizer [59] with a learning rate decreasing exponentially from $10^{-2}$ to $10^{-4}$. We performed 200 training steps with batch size 32.

The first classifier $\widetilde{Q}(y_1, y_2|x_t)$ was a convolutional network consisting of 2 convolutional layers with 64 and 96 channels and three hidden layers with 512, 256 and 256 units. This classifier is time-dependent and used 128-dimensional Gaussian random features to embed the time. The output was a 3x3 matrix encoding the predicted log-probabilities. The classifier was trained on trajectories sampled from the reverse SDE of the base diffusion models using the AdaDelta optimizer [67] with a learning rate of 1.0. We performed 700 training steps with batch size 128. For the first 100 training steps the classifier was only trained on terminal samples.

The second conditional classifier $\widetilde{Q}(y_3|y_1, y_2, x_t)$ was a similar convolutional network with 2 convolutional layers with 64 channels and two hidden layers with 256 units. This classifier is conditioned both on time and on $(y_1, y_2)$. The time variable was embedded used 128-dimensional Gaussian random features. The $(y_1, y_2)$ variables were encoded using a 1-hot encoding scheme. The output of the classifier was the three predicted log-probabilities for $y_3$. Contrary to the first classifier, this one was not trained on the base diffusion models but rather on samples from the posterior $\widetilde{p}(x|y_1, y_2)$. It's loss function was:

$$\mathcal{L}_c(\phi) = \mathop{\mathbb{E}}_{(\hat{x}_0, \hat{x}_t, \hat{y}_2, \hat{y}_1, t) \sim \widetilde{p}(x_0, x_t|y_1, y_2)\widetilde{p}(y_1)\widetilde{p}(y_2)p(t)} \left[ \sum_{\hat{y}_3=1}^{m} -w_{\hat{y}_3}(\hat{x}_0) \log \widetilde{Q}_\phi(\hat{y}_3|\hat{y}_1, \hat{y}_2, \hat{x}_t) \right] \quad (116)$$

where $w_{\hat{y}_3}(\hat{x}_0)$ is estimated using the first classifier. The classifier was trained using the AdaDelta optimizer [67] with a learning rate of 0.1. We performed 200 training steps with batch size 128.

**Sampling.** Sampling from both the individual base models and the composition was done using the Predictor-Corrector sampler [34]. We performed sampling over 500 time steps to generate the

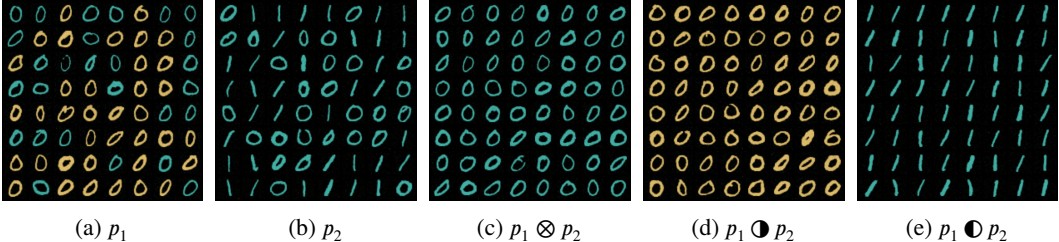

|  (a) $p_1$ | (b) $p_2$ | (c) $p_1 \otimes p_2$ | (d) $p_1 \ominus p_2$ | (e) $p_1 \ominus p_2$ |

Figure G.1: **Diffusion model composition on colored MNIST.** (a,b) samples from base diffusion models. (c-e) samples from the resulting harmonic mean and contrast compositions.

samples shown in Figure 5. The samples used to train the classifier were generated using the same method.

When sampling from the composition we found that using scaling for the classifier guidance was generally necessary to achieve high-quality results. Without scaling, the norm of the gradient over the first and second classifier was too small relative to the gradient predicted by the score function, and hence did not sufficiently steer the mixture towards samples from the posterior. Experimentally, we found that scaling factor 10 for the first classifier and scaling factor 75 for the second produced high quality results.

# G    Additional Results

## G.1    Binary Operations for MNIST Digit Generation via Diffusion Models

Here we present a variant of the colored digit generation experiment from Section 6 ("Colored MNIST generation via diffusion models"). using 2 diffusion models. This allows us to better illustrate the harmonic mean and contrast operations on this image domain. In a similar fashion to the experiment in Section 6 ("Colored MNIST generation via diffusion models"), we trained two diffusion models to generate colored MNIST digits. $p_1$ was trained to generate red and green 0 digits and $p_2$ was trained to generate green 0 and 1 digits. As before, we used post-processing to map green to cyan and red to beige.

**Implementation details.**    The diffusion models used in this experiment and their training procedure were exactly the same as in Section F.3. The sampling method used to obtain samples from the base models and their compositions was also the same. We found that scaling the classifier guidance was generally required for high-quality results, and used a scaling factor of 20 in this experiment.

The classifier was a convolutional network with 2 convolutional layers consisting of 32 and 64 channels and two hidden layers with 512 and 128 units. The classifier's time input was embedded using 128-dimensional Gaussian random features. The output was a 2x2 matrix encoding the predicted log-probabilities $\widetilde{Q}(y_1, y_2 \mid x_t)$. The classifier was trained on trajectories, sampled from the reverse SDE of the base diffusion models, using the AdaDelta optimizer [67] with a learning rate of 0.1 and a decay rate of 0.97. We performed 200 training steps with batch size 128. For the first 100 training steps, the classifier was only trained on terminal samples.

**Results.**    Figure G.1 shows samples obtained from the two trained diffusion models $p_1$, $p_2$ and from the harmonic mean and contrast compositions of these models. We observe that the harmonic mean generates only cyan zero digits, because this is the only type of digit on which both $p_1$ and $p_2$ have high density. The contrast $p_1 \ominus p_2$ generates beige zero digits from $p_1$. However, unlike $p_1$, it does not generate cyan zero digits, as $p_2$ has high density there. The story is similar for $p_1 \ominus p_2$, which generates cyan one digits from $p_2$, but not zero digits due to $p_1$ having high density over those.

## G.2    MNIST Subset Generation via Diffusion Models

We report in this section on additional results for composing diffusion models on the standard MNIST dataset. We trained two diffusion models: $p_{\{0,\dots,5\}}$ was trained to generate MNIST digits 0 through 5,

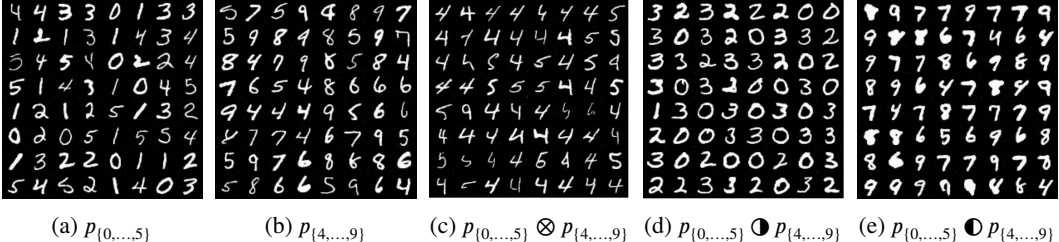

(a) $p_{\{0,\dots,5\}}$    (b) $p_{\{4,\dots,9\}}$    (c) $p_{\{0,\dots,5\}} \otimes p_{\{4,\dots,9\}}$    (d) $p_{\{0,\dots,5\}} \mathbin{\pmb{\mathbb{O}}} p_{\{4,\dots,9\}}$    (e) $p_{\{0,\dots,5\}} \mathbin{\pmb{\mathbb{O}}} p_{\{4,\dots,9\}}$

Figure G.2: **Diffusion model composition on MNIST.** (a,b) samples from base diffusion models. (c-e) samples from the resulting harmonic mean and contrast compositions.

and $p_{\{4,\dots,9\}}$ to generate MNIST digits 4 through 9. The training procedure and models used in this experiment were the same as in Section G.1.

Figure G.2 shows samples obtained from the two diffusion models, from the harmonic mean, and from the contrast compositions of these models. We observe that the harmonic mean correctly generates mostly images on which both diffusion models' sampling distributions have high probability, i.e. digits 4 and 5. For the contrasts we see that in both cases digits are generated that have high probability under one model but low probability under the other. We observe some errors, namely some 9's being generated by the harmonic mean and some 4's being generated by the contrast $p_{\{0,\dots,5\}} \mathbin{\pmb{\mathbb{O}}} p_{\{4,\dots,9\}}$. This is likely because 4 and 9 are visually similar, causing the guiding classifier to misclassify them, and generate them under the wrong composition.

We also present binary operations between three distributions. In Figure G.3 and G.4, $p_0$ models even digits, $p_1$ models odd digits, and $p_2$ models digits that are divisible by 3. We color digits $\{0,6\}$ purple, $\{3,9\}$ blue, $\{4,8\}$ orange, and $\{1,5,7\}$ beige. In Figure G.3, harmonic mean of $p_0$ and $p_2$ generates the digit 0 and 6, whereas the contrast of $p_0$ with $p_2$ shows even digits non-divisible by 3 ($p_0 \mathbin{\pmb{\mathbb{O}}} p_2 = \{4,8\}$), and odd numbers that are divisible by 3 ($p_0 \mathbin{\pmb{\mathbb{O}}} p_2 = \{3,9\}$). We observe that the samples from $p_0 \otimes p_2$ inherit artifacts from the base generator for $p_2$ (the thin digit 0), which shows the impact that the base models have on the composite distribution. In Figure G.4 we present similar results between odd digits ($\{5,3,5,7,9\}$). We noticed that samples from both $p_1 \mathbin{\pmb{\mathbb{O}}} p_2$ and $p_1 \mathbin{\pmb{\mathbb{O}}} p_2$ includes a small number of the digit 3.

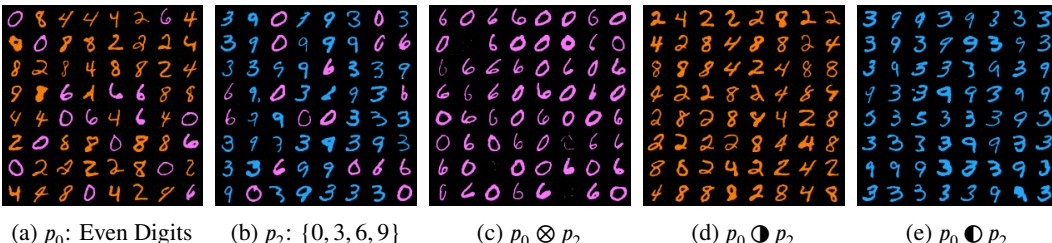

(a) $p_0$: Even Digits    (b) $p_2$: $\{0,3,6,9\}$    (c) $p_0 \otimes p_2$    (d) $p_0 \mathbin{\pmb{\mathbb{O}}} p_2$    (e) $p_0 \mathbin{\pmb{\mathbb{O}}} p_2$

Figure G.3: **Composing even digits and multiples of three.** (a,b) samples from base diffusion models. (c-e) samples from the resulting harmonic mean and contrast compositions.

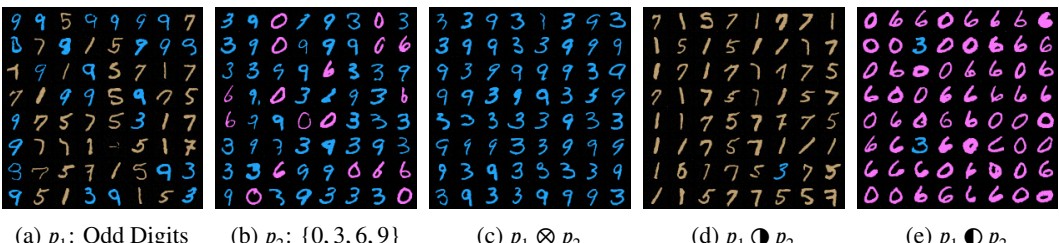

(a) $p_1$: Odd Digits    (b) $p_2$: $\{0,3,6,9\}$    (c) $p_1 \otimes p_2$    (d) $p_1 \mathbin{\pmb{\mathbb{O}}} p_2$    (e) $p_1 \mathbin{\pmb{\mathbb{O}}} p_2$

Figure G.4: **Composing odd digits and multiples of three.** (a,b) samples from base diffusion models. (c-e) samples from the resulting harmonic mean and contrast compositions.

### G.3 Chaining: Sequential Composition of Multiple Distributions

We present results on chaining binary composition operations sequentially on a custom colored MNIST dataset.

**Setup.** We start with three base generative models that are trained to generate $p_1$, $p_2$ and $p_3$ in Figure G.5. Specifically, $p_1$ is a uniform distribution over digits $\{0, 1, 2, 3, 4, 5\}$, $p_2$ is a uniform distribution over even digits $\{0, 2, 4, 6, 8\}$, and $p_3$ is a uniform distribution over digits divisible by 3: $\{0, 3, 6, 9\}$. Note that we use a different color for each digit consistent across $p_1, p_2, p_3$. Our goal is to produce combinations of chained binary operations involving all three distributions, where two of them were combined first, then in a second step, combined with the third distribution through either harmonic mean $\otimes$ or contrast $\mathbb{O}$.

**Binary Classifier Training.** Consider, for example, the operation $(p_1 \otimes p_2) \mathbb{O} p_3$. We use the same classifier training procedure for $p_1$ versus $p_2$, as well as the composite model $(p_1 \otimes p_2)$ versus $p_3$, except that in the later case we sample from composite model as a whole. Our classifier training simultaneously optimizes the terminal classifier and the intermediate state classifier.

**Implementation Detains.** We adapted diffusion model training code for the base distributions from [33]. Our diffusion model used a UNet backbone with four latent layers on both the contracting and the expansive path. The contracting channel dimensions were $[64, 128, 256, 256]$ with the kernel size 3, and strides $[1, 2, 2, 2]$. The time embedding used a mixture of 128 sine and cosine feature pairs, with a total of 256 dimensions. These features were passed through a sigmoid activation and then expanded using a different linear head for each layer. The activations were then added to the 2D feature maps at each layer channel-wise. We used a fixed learning rate of 0.01 with Adam optimizer [59].

We adapted the classifier training code from the MNIST example in Pytorch [68]. Our binary classifier has two latent layers with channel dimensions $[32, 64]$, stride 1 and kernel width of 3. We use dropout on both layers: $p_1 = 25\%, p_2 = 50\%$. We train the network for 200 epochs on data sampled online in batches of 256 from each source model, and treat the composite model in the second step the same way. We use the Adadelta [67] optimizer with the default setting of learning rate 1.0.

**Sampling.** We generate samples according to Appendix A.2 and multiply the gradient by $\alpha = 20$.

**Results.** Row 3 from Figure G.5 contains mostly zeros with a few exceptions. This is in agreement with harmonic mean being symmetric. In $p_1 \otimes (p_2 \otimes p_3)$, digits that are outside of the intersection still appear with thin strokes.

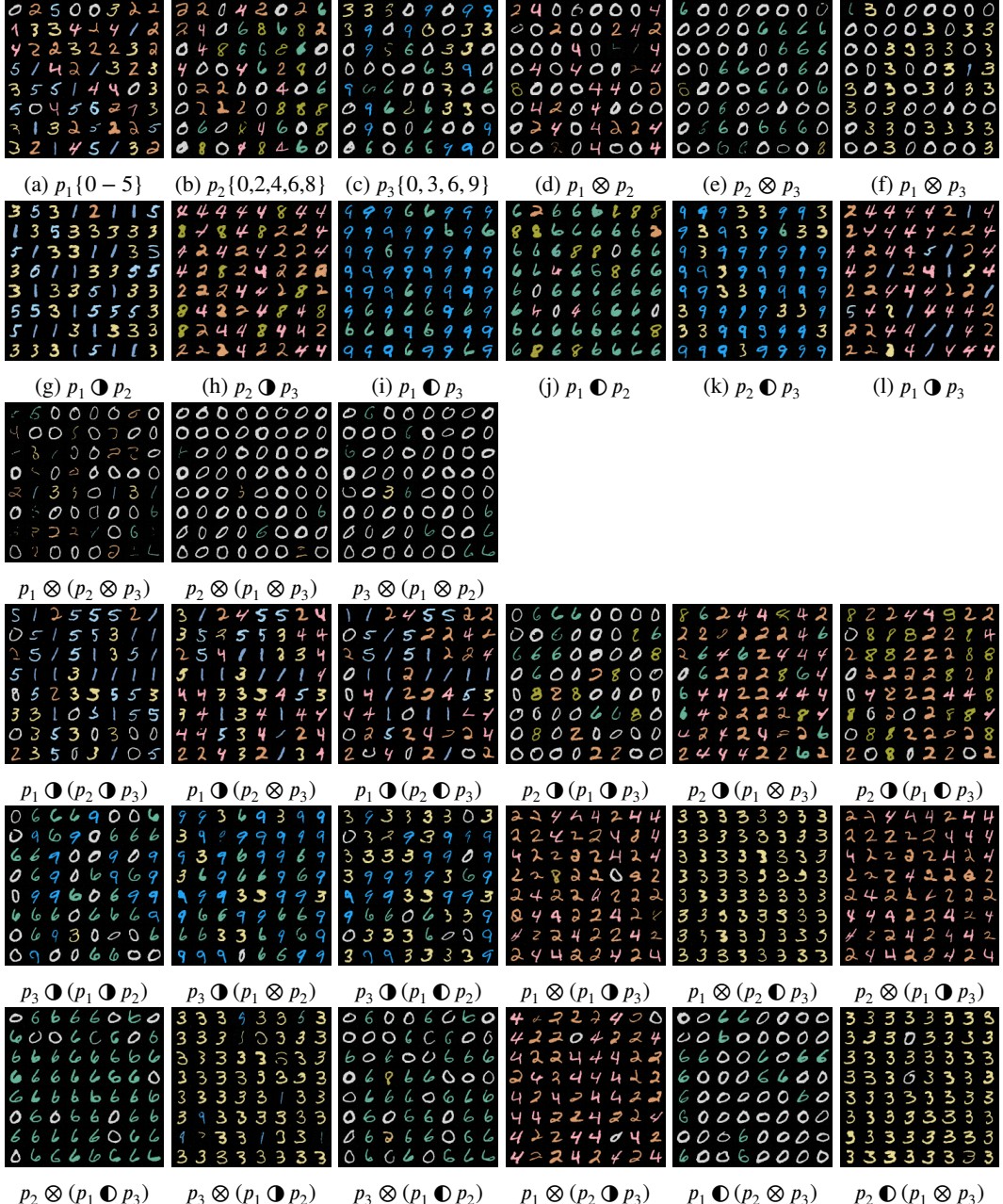

Figure G.5: **Chaining Binary Operations.** (a-c) Samples from 3 pre-trained diffusion models. (d-l) Samples from binary compositions. (row 3) The harmonic mean between all three. (row 4 and beyond) various ways to chain the operations. Parentheses indicate the order of composition.

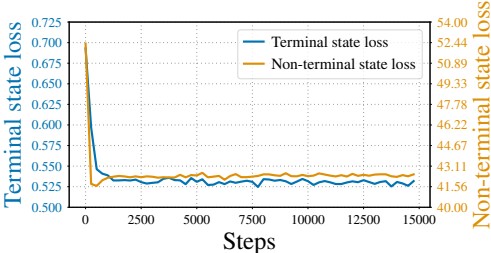
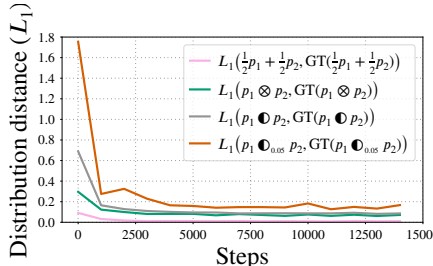

Figure G.6: Training curves of the classifier $\widetilde{Q}_\phi(y_1, y_2|\cdot)$ in GFlowNet 2D grid domain. The experimental setup corresponds to Section 6 ("2D distributions via GFlowNet") and Figure 2 (top row). **Left**: Terminal state loss and non-terminal state loss (as defined in Algorithm A.1) as functions of the number of training steps. **Right**: $L_1$ distance between learned distributions (compositions obtained through classifier-based mixture and guidance) and ground-truth composition distributions as the function of the number of training steps. $L_1(p, q) = \sum_{x \in \mathcal{X}} |p(x) - q(x)|$.

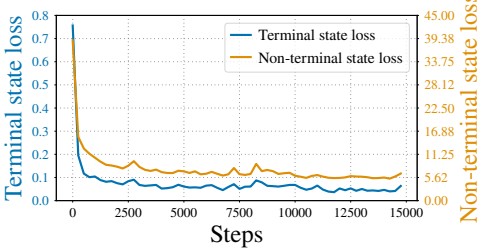

Figure G.7: Training curves of the classifier $\widetilde{Q}_\phi(y_1, y_2|\cdot)$ in GFlowNet molecule generation domain. The experimental setup corresponds to Section 6 ("Molecule generation via GFlowNet") and Figure 3 (a-c). The curves show terminal state loss and non-terminal state loss (as defined in Algorithm A.1) as functions of the number of training steps.

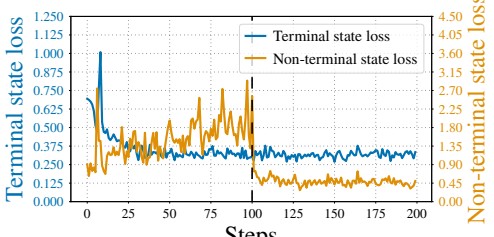

Figure G.8: Training curves of the classifier $\widetilde{Q}_\phi(y_1, y_2|\cdot)$ in diffusion MNIST image generation domain. The experimental setup corresponds to Section G.1 and Figure G.1. The curves show terminal state loss and non-terminal state loss (as defined in equations (26), (28)) as functions of the number of training steps. The non-terminal loss optimization begins after the first 100 training steps (shown by the black dashed line).

## G.4 Classifier Learning Curves and Training Time

We empirically evaluated classifier training time and learning curves. The results are shown in Figures G.6, G.7, G.8 and Tables G.1, G.2.

Figures G.6, G.7, G.8 show the cross-entropy loss of the classifier for terminal (16) and non-terminal states (18) as a function of the number of training steps for the GFlowNet 2D grid domain, the molecular generation domain, and the Colored MNIST digits domain respectively. They show that the loss drops quickly but remains above 0. Figure G.6 further shows the distances between the learned compositions and the ground truth distributions as a function of the number of training steps of the classifier. For all compositions, as the classifier training progresses, the distance to the ground truth distribution decreases. Compared to the distance at initialization we observe almost an order of magnitude distance reduction by the end of the training.

The runtime of classifier training is shown in Tables G.1 and G.2. We report the total runtime, as well as separate measurements for the time spent sampling trajectories and training the classifier. The classifier training time is comparable to the base generative model training time. However, most of the classifier training time (more than 70%, or even 90%) was spent on sampling trajectories from base generative models. Our implementation of the training could be improved in this regard, e.g. by sampling a smaller set of trajectories once and re-using trajectories for training and by reducing the number of training steps (the loss curves in Figures G.6, G.7, G.8 show that classification losses plateau quickly).

Table G.1: Summary of base GFlowNet and classifier training time in molecule generation domain. The experimental setup corresponds to Section 6 ("Molecule generation via GFlowNet") and Figure 3 (a-c). All models were trained with a single GeForce RTX 2080 Ti GPU.

| | |
|---|---|
| Base GFlowNet training steps | 20 000 |
| Base GFlowNet batch size | 64 |
| Base GFlowNet training elapsed real time | 6h 47m 11s |
| Classifier training steps | 15 000 |
| Classifier batch size | 8 trajectories per base model (all states used) |
| Classifier training total elapsed real time | 9h 2m 19s |
| Classifier training data generation time | 6h 35m 58s (73%) |

Table G.2: Summary of base diffusion and classifier training time in MNIST image generation domain. The experimental setup corresponds to Section G.1 and Figure G.1. All models were trained with a single Tesla V100 GPU.

| | |
|---|---|
| Base diffusion training steps | 200 |
| Base diffusion batch size | 32 |
| Base diffusion training elapsed real time | 10m 6s |
| Classifier training steps | 200 |
| Classifier batch size | 128 trajectories per base model (35 time-steps per trajectory) |
| Classifier training total elapsed real time | 30m 12s |
| Classifier training data generation time | 29m 22s (97%) |

Table G.3: Average pairwise similarity [36] of molecules generated by GFlowNets trained on 'SEH', 'SA', 'QED' rewards at different values of $\beta$. For each combination (reward, $\beta$) a GFlowNet was trained with the corresponding reward $R(x)^\beta$. Then, 5 000 molecules were generated. The numbers in the table reflect the average pairwise Tanimoto similarity of top 1 000 molecules (selected according to the target reward function).

| | SEH | SA | QED |
|---|---|---|---|
| $\beta = 1$ | 0.527 | 0.539 | 0.480 |
| $\beta = 4$ | 0.529 | 0.527 | 0.464 |
| $\beta = 10$ | 0.535 | 0.500 | 0.438 |
| $\beta = 16$ | 0.548 | 0.465 | 0.422 |
| $\beta = 32$ | 0.585 | 0.411 | 0.398 |
| $\beta = 96$ | 0.618 | 0.358 | 0.404 |

Table G.4: Number of Tanimoto-separated modes found above reward threshold. For each combination (reward, $\beta$) a GFlowNet was trained with the corresponding reward $R(x)^\beta$, and then 5 000 molecules were generated. Cell format is "$A/B$", where $A$ is the number of Tanimoto-separated modes found above the reward threshold, and $B$ is the total number of generated molecules above the threshold. Analogously to Figure 14 in [36], we consider having found a new mode representative when a new molecule has Tanimoto similarity smaller than 0.7 to every previously found mode's representative molecule. Reward thresholds (in [0, 1], normalized values) are 'SEH': 0.875, 'SA': 0.75, 'QED': 0.75. Note that the normalized threshold of 0.875 for 'SEH' corresponds to the unnormalized threshold of 7 used in [36].

| | SEH | SA | QED |
|---|---|---|---|
| $\beta = 1$ | 15 / 17 | 37 / 37 | 0 / 0 |
| $\beta = 4$ | 12 / 17 | 82 / 82 | 0 / 0 |
| $\beta = 10$ | 85 / 109 | 332 / 337 | 18 / 18 |
| $\beta = 16$ | 190 / 280 | 886 / 910 | 253 / 253 |
| $\beta = 32$ | 992 / 1821 | 2859 / 3080 | 3067 / 3124 |
| $\beta = 96$ | 1619 / 4609 | 4268 / 4983 | 4470 / 4980 |

## G.5 Analysis of Sample Diversity of Base GFlowNets in Molecule Generation Domain

In order to assess the effect of the reward exponent $\beta$ on mode coverage and sample diversity, we evaluated samples generated from GFlowNets pre-trained with different values of $\beta$. The results are in Tables G.3 and G.4. The details of the evaluation and the reported metrics are described in the table captions. As expected, larger reward exponents shift the learned distributions towards high-scoring molecules (the total number of molecules with scores above the threshold increases). For 'SA' and 'QED' models we don't observe negative effects of large $\beta$ on sample diversity and mode coverage: the average pairwise similarity of top 1 000 molecules doesn't grow as $\beta$ increases and the ratio of Tanimoto-separated modes remains high. For 'SEH' models we observe a gradual increase in the average pairwise similarity of top 1 000 molecules and a gradual decrease in the ratio of Tanimoto-separated modes. However, the total number of separated modes grows as $\beta$ increases, which indicates that larger reward exponents do not lead to mode dropping.

## G.6   Summary of Pairwise Distribution Distances in Molecule Generation Domain

Table G.5: Estimated pairwise earth mover's distances between distributions shown in Table 1.

| | y=SEH | y=SA | y=QED | y=SEH,SA | y=SEH,QED | y=SA,QED | y=SEH,SA,QED | y=SEH × 3 | y=SA × 3 | y=QED × 3 |
|---|---|---|---|---|---|---|---|---|---|---|
| y=SEH | 0 | 4.42 | 5.77 | 3.39 | 4.20 | 4.88 | 4.10 | 2.46 | 4.44 | 5.73 |
| y=SA | 4.42 | 0 | 5.88 | 3.26 | 5.15 | 4.59 | 4.20 | 4.39 | 2.55 | 5.89 |
| y=QED | 5.77 | 5.88 | 0 | 5.40 | 4.02 | 3.85 | 4.20 | 5.80 | 5.90 | 3.20 |
| y=SEH,SA | 3.39 | 3.26 | 5.40 | 0 | 4.25 | 4.19 | 3.68 | 3.39 | 3.30 | 5.39 |
| y=SEH,QED | 4.20 | 5.15 | 4.02 | 4.25 | 0 | 3.80 | 3.67 | 4.22 | 5.19 | 4.00 |
| y=SA,QED | 4.88 | 4.59 | 3.85 | 4.19 | 3.80 | 0 | 3.65 | 4.91 | 4.59 | 3.87 |
| y=SEH,SA,QED | 4.10 | 4.20 | 4.20 | 3.68 | 3.67 | 3.65 | 0 | 4.12 | 4.23 | 4.20 |
| y=SEH × 3 | 2.46 | 4.39 | 5.80 | 3.39 | 4.22 | 4.91 | 4.12 | 0 | 4.43 | 5.73 |
| y=SA × 3 | 4.44 | 2.55 | 5.90 | 3.30 | 5.19 | 4.59 | 4.23 | 4.43 | 0 | 5.90 |
| y=QED × 3 | 5.73 | 5.89 | 3.20 | 5.39 | 4.00 | 3.87 | 4.20 | 5.73 | 5.90 | 0 |

