# OpenReview forum: "Compositional Sculpting of Iterative Generative Processes"
_NeurIPS.cc/2023/Conference — NeurIPS 2023 poster_

### Official Review · Reviewer_m7vZ · 2023-06-23

**Soundness:** 3 good
**Presentation:** 2 fair
**Contribution:** 3 good
**Rating:** 6
**Confidence:** 1

**Summary:**

This paper proposes an approach of Compositional Sculpting for iterative generative models, including GFlowNets and defusion models.
The model uses classifier guidance to sample from the target posterior distribution composed of pre-trained base models.
The paper also proposes a training algorithm for the classifier.
The approach is validated by empirical analyses of an image dataset and molecular generation.

**Strengths:**

- The method is general enough for different GFlowNets and defusion models.

- The empirical results validate the method.

**Weaknesses:**

(1) It would be convincing to have more complicated experiments, especially for image data.
Colored MNIST might be too small and simple.

(2) More clarity might be helpful for the following points.

(2A) What is the model in line 193? "Under the model we have introduced the variables y_1, ... y_n are dependent given a state s in S, but, are independent given a terminal state x in X."

(2B) I am confused while reading line 209. The sampling scheme is 1) sample y from its prior 2) sample tau given y. But the following sentence says sampling y given tau.

**Questions:**

Please refer to the weakness section.

Typos - Line 217: train train

**Limitations:**

The paper has a limitation section.
It seems not to have a dedicated section for social impact.

---

> ### Author Rebuttal · Authors · 2023-08-09
>
> Thank you for your review of our paper and your feedback. We’ve provided some clarification in response to your questions below.
>
> > What is the model on line 193?
>
> Equations (2)-(4) constitute a graphical model over the variables $x$ and $y_1, \dots, y_n$. We introduce a specialization of this model for GFlowNets in the first paragraph of section 3.1. This is the model we refer to on line 193 in section 3.2. We will clarify in the text that this refers specifically to the specialized model for GFlowNets.
>
> > Confused when reading line 209. The sampling scheme is 1) sample y from its prior 2) sample tau given y. But the following sentence says sampling y given tau.
>
> We allow conditioning on multiple observations. Thus, there are multiple i.i.d. variables $y_1, \dots y_n$, one for each observation. Line 209 explains that when generating samples to train on, the first observation $\widehat y_1$ is sampled from its prior, $\widehat \tau$ is sampled given $\widehat y_1$, and the remaining observations $\widehat y_2, \dots \widehat y_n$ are then sampled given $\widehat{\tau}$.
>
> > It seems not to have a dedicated section for social impact.
>
> Most societal impact issues are common to other work on generative models. However, our work has a positive impact in terms of opening the door towards efficient compositions of existing models, enabling wider applicability and computational savings. We will expand on this in a dedicated paragraph in the conclusion.

---

> > ### Comment · Reviewer_m7vZ · 2023-08-21
> >
> > Thank you for the rebuttal. I raised the score.

---

> ### Comment · Area_Chair_zygR · 2023-08-18
> **Adjust score after rebuttal**
>
>  Dear Reviewer m7vz:
>
> Thanks for your review. Please read the rebuttal and start discussing points of disagreement raised by the authors and you. Most reviewers are towards accepting the paper while you are relatively negative. Please reconsider your score and adjust it accordingly.
>
> AC

---

### Official Review · Reviewer_Ean9 · 2023-07-04

**Soundness:** 4 excellent
**Presentation:** 4 excellent
**Contribution:** 3 good
**Rating:** 6
**Confidence:** 5

**Summary:**

The paper describes a way in which, given sequential samplers from multiple probability distributions, a combination of the samplers can be used to sample from a composition of the distributions. To be precise, the sequential samplers are either GFlowNets or diffusion models, the combination of samplers is a weighted combination of action distributions at each intermediate sampling step, and the composition of distributions can be defined by simple soft conjunction and set difference operators. This is demonstrated in toy illustratory experiments and multiobjective molecule synthesis (GFlowNet) and MNIST with digit class and colour attributes (diffusion).

**Strengths:**

From the perspective of someone who works on both GFlowNets and diffusion models, this is a very well-written paper.
- The text reads naturally and the right amount of detail is given in the main text. There is a good choice of illustrations to help the reader.
- The composition of multiple GFlowNets has not been considered before and could be useful, especially in multiobjective problems. The unifying perspective on classifier guidance is also an advantage (but see below).
- Code is provided, a nice addition to the paper.
- I checked the GFlowNet-related math and believe it to be sound.

**Weaknesses:**

- Line 234, 255, 634, 643, maybe others: typo "GFLowNet"
- It would be good to explain why / state as a subclaim that (8) is a policy (i.e., sums to 1 over $s'$), which is not actually obvious from the definition.
  - It relies on the fact that $p(y|s) = \sum_{s'}p(y|s')p(s'|s)$, which follows from conditional independence of $y$ (a function of the final state) and $s$ given $s'$. That is a consequence of the Markov property in GFlowNets. A note should be made about this.
  - The equality may not actually hold in practice, when $p(y|-)$ is a trained classifier, so (8) may not exactly sum to 1. What do you do in this case (in the experiments)?
- The results on molecule generation raise a few questions:
  - The reward exponent was set to $\beta=32$ or $\beta=96$, which is far larger than in past work, where it was at most 16. Why was such a choice made? This is suspicious, since convergence and mode collapse issues worsen at low temperatures.
  - Related, with such high exponents, one wonders about mode coverage in the learned distributions. Have you considered the in-sample diversity of the generated molecules (e.g., as measured by average Tanimoto similarity or  diverse top-k metrics)?
- On related work:
  - There is no substantial discussion of related work in the main text, even though there is a large body of work on compositional generation and classifier guidance with diffusion models (e.g., the many papers cited in the second paragraph of the introduction).
  - In the Appendix, the connection with [23] is discussed. The proposed method can easily turn a collection of classifiers for different objectives into a classifier for any convex combination of the objectives. It would be interesting to empirically compare this with conditioning of the model on the linear scalarization weights used in [23].
  - The paper would be stronger if more explicit unifying connections were made between guidance in diffusion models and in GFlowNets. Note that diffusion models in a fixed time discretization are actually GFlowNets of a certain structure (cf. "A theory of continuous generative flow networks" [arXiv:2301.12594, ICML 2023] and "Unifying generative models with GFlowNets and beyond" [arXiv:2209.02606]). Classifier guidance using the gradient of $p_t(y|x_t)$ should be the continuous-time limit of equation (8).

**Questions:**

Please see "weaknesses" above.


**Limitations:**

Yes

---

> ### Author Rebuttal · Authors · 2023-08-09
>
> Thank you for the thorough review of our paper and thoughtful feedback. Please find our response to the raised questions below.
>
> > It would be good to explain why / state as a subclaim that (8) is a policy (i.e., sums to 1 over $s’$).
>
> Following your suggestion, in the subsequent revision of the paper we will include a note explaining why (8) is a valid policy (sums to 1 over $s’$). The preliminary text of the note including your suggested proof sketch is below:
>
> ```
> Note that (8) is a valid forward policy, i.e. the distribution sums up to 1 over $s'$. This property follows from the relationship $p(y \vert s) = \sum_{s'} p(y \vert s') p_F(s' \vert s)$ which is implied by the probabilistic model: $y$ is a (stochastic) function of the terminal state $x$, $y$ and $s$ are independent given $s'$.
> ```
>
> We will provide more detailed proof in the appendix.
>
> > The equality may not actually hold in practice. What do you do in this case (in the experiments)?
>
> It is true that in practice, when only an approximation of the classifier is available, the policy constructed according to eq (8) may not exactly sum to 1. In our experiments, we expressed the conditional policy in terms of log-probabilities (logits) and computed the probabilities as $\operatorname{softmax}_{s’}(\log p_F(s’ \vert s) + \log p(y \vert s’) - \log p (y \vert s))$, where the softmax operation ensures that the obtained distribution sums up to 1 over $s’$. In theory (assuming the perfectly learned classifier), the softmax operation can be replaced with simple element-wise exponentiation, but using the softmax is also correct.
>
> > The reward exponent was set to $\beta = 32$ or $\beta = 64$. Why was such a choice made?
>
> We chose to set the reward exponent $\beta$ to $32$ and $96$ due to the following reasons:
> * $\beta = 96$ was used in “Multi-objective GFlownets” [23] for a similar task (c.f. parameters in Table 12, Section D.4 of [23])
> * composition of models concentrated on high-scoring molecules is a more challenging and application-relevant task
>
> > Have you considered the in-sample diversity of the generated molecules
>
> We evaluated samples generated from GFlowNets pre-trained with different reward exponents $\beta$, in order to assess the effect of $\beta$ on mode coverage and sample diversity. The results are in Tables R.3 and R.4 in the rebuttal PDF. The details of the evaluation and the reported metrics are described in the table captions. As expected, larger reward exponents shift the learned distributions towards high-scoring molecules (the total number of molecules with scores above the threshold increases). For ‘SA’ and ‘QED’ models we don’t observe negative effects of large $\beta$ on sample diversity and mode coverage: the average pairwise similarity of top 1000 molecules doesn’t grow as $\beta$ increases and the ratio of Tanimoto-separated modes remains high. For ‘SEH’ models we observe a gradual increase in the average pairwise similarity of the top 1000 molecules and a gradual decrease in the ratio of Tanimoto-separated modes. However, the total number of separated modes grows as $\beta$ increases, which indicates that larger reward exponents don’t lead to mode dropping.
>
> > There is no substantial discussion of related work in the main text. The paper would be stronger if more explicit unifying connections were made.
>
> We plan to utilize the additional content page to expand the discussion of the related work on compositional generation and guidance in diffusion models. In particular, the papers cited in the introduction. We also will make a note that would help a reader to better position our work in the view of unifying connections between guidance in diffusion models and GFlowNets (and continuous GFlowNets).
>
> > The proposed method can easily turn a collection of classifiers for different objectives into a classifier for any convex combination of the objectives. It would be interesting to empirically compare with conditioning of the model on the linear scalarization weights used in [23]
>
> We understand the first part of the question in the following way: “The proposed method can easily turn a collection of GFlowNets for different objectives into a GFlowNet for any convex combination of the objectives”. Please let us know if our understanding is correct.
>
> Convex combination of the rewards $R(x | w) = \sum_i w_i R_i(x) = \sum_i w_i Z_i  p_i(x)$ corresponds to a mixture distribution $p(x | w) =\sum_i \left(  \frac{w_i Z_i}{\sum_j w_j Z_j}  p_i(x) \right)$ which can be realized by both Multi-objective GFlowNets [23] and our approach.
>
> We mainly focus on harmonic mean, contrast, and other compositions beyond mixtures. Note that in our scenario (access to forward policies of pre-trained GFlowNets), sampling from the mixture can be realized without training a classifier: it is sufficient to sample an index of the base model from a categorical distribution, and then run the forward process of the selected base GFlowNet. In our approach, we represent the mixture as an individual GFlowNet forward policy (expressed through a classifier) because at the next step, we apply classifier guidance to this policy.

---

> > ### Comment · Reviewer_Ean9 · 2023-08-14
> > **Response**
> >
> > Thank you for the answers! I have no further questions and maintain my positive assessment of the paper.
> >
> > > We understand the first part of the question in the following way: “The proposed method can easily turn a collection of GFlowNets for different objectives into a GFlowNet for any convex combination of the objectives”. Please let us know if our understanding is correct.
> >
> > Correct, my mistake.

---

### Official Review · Reviewer_eEwT · 2023-07-05

**Soundness:** 2 fair
**Presentation:** 3 good
**Contribution:** 2 fair
**Rating:** 6
**Confidence:** 3

**Summary:**

The paper studies the problem of composing independently trained generative processes of diffusion-based generative models and GFlowNets. The paper considers a setting where one has access to $m$ pre-trained samplers for $\{p_i(x)\}_{i=1}^m$, and the goal is to obtain a sampler which corresponds to a composition of these processes. Specifically, the authors consider two ways of composing the processes, namely harmonic mean: where the likelihood of the composition is high only where the component processes have high likelihood and contrast The authors frame this as sampling from a conditional distribution $p(x|y)$, where y is an observation which denotes the index of the process a sample is generated from. This results in a procedure analogous to classifier guidance which is popular in the diffusion literature. The authors show how the classifier guidance results in sampling from the desired composition. A critical component of this procedure is learning the classifier $p(y|s)$, for which the authors propose a MLE based procedure using trajectories from the base model. The authors first validate their approach on some synthetic tasks on a 2D grid, followed by experiments on small molecule generation with GFlowNets and colored MNIST with diffusion.

**Strengths:**

* The paper studies an interesting question - which is relevant to the community. In particular, methods to leverage pre-trained models for various downstream tasks are becoming increasingly important with the growing adoption of pre-trained models for various domains.
* Since similar classifier guidance approaches have been studied extensively in the literature on diffusion models, the novelty is relatively limited. Nonetheless, there are several technical aspects of the approach such as the classifier training scheme that are novel (to the best of my knowledge).
* The proposed method is relatively simple conceptually, and in terms of implementation.
* The experiments are well designed, and the results are quite promising, albeit with some caveats I mention below.
* The paper overall is quite well written and easy to follow. I also appreciate the authors including the code with the submission.

**Weaknesses:**

* As the authors discuss in Section 3, their theoretical analysis is analogous to classifier guidance in diffusion models. On the other hand, [1] establishes equivalence between GFlowNets and diffusion models. As a results, it seems to me that the insights provided by the theoretical analysis aren’t particularly novel even though the path to achieving them was different (which could be seen as useful on it’s own)
* The experiments on diffusion are limited to a simple coloured MNIST task, with no baselines from the classifier-guided diffusion literature.
* The central aspect of the approach is learning the classifiers, however there is no analysis on the classifiers - e.g. how accurate are the classifiers? what is the effect of the classifier performance on the results of the composition? How does the training of the classifier compare to simply training a GFlowNet from scratch in terms of runtime?
* (Minor) A simple baseline that is missing in the experiments is training a GFlowNet with the appropriate composition from scratch.

[1] Unifying Generative Models with GFlowNets and Beyond. Zhang et al. 2022. arXiv:2209.02606

**Questions:**

* Could you please address the questions about the analysis of the effect of the classifier along with some runtime details?
* What are the challenges in applying the method to more complicated tasks with diffusion models?
* One natural question I had after reading the paper is that since the method encompasses GFlowNets and diffusion models, is it conceivable to be able to compose a mixture of GFlowNets and diffusion models on mixed continuous/discrete tasks?
* In the experiments, the authors consider a maximum of 3 models being composed. Is there a reason to be limited to composing only 3 models? If not at least in the 2D example an experiment with more models might be helpful.

Minor:
* The main PDF on OpenReview appears to have some old version of the Appendix at the end. I assume that was by mistake?
* In several places there is inconsistent usage of “GFLowNet” in place of “GFlowNet”.

**Limitations:**

The authors already discuss some limitations of the approach - effect of the classifier as well as the quality of the underlying models. I would also add limited evaluation and lack of comparison to existing approaches.

---

> ### Author Rebuttal · Authors · 2023-08-09
>
> Thank you for your review of our paper and your insightful feedback. We have addressed the main questions and concerns you have raised below.
>
> > Classifier guidance for GFlowNets is not too novel considering the equivalence to diffusion models.
>
> This is a fair point. However, despite the known connection, classifier guidance for GFlowNets has not been proposed in prior work as far as we know. Our focus is on efficiently generating samples from compositions of iterative generative models using classifier guidance, which necessitated introducing classifier guidance for GFlowNets.
>
> > No analysis of the classifiers. How accurate are they, how does accuracy affect performance?
>
> The quality of the classifier is fundamental to the method, as the classifier guides the generative process of the mixture of base models $p_i(s)$ towards sampling from the posterior $\tilde p(s | y_1, \dots, y_n)$. If the classifier is poor, the sampling distribution will not match $\tilde p(x | y_1, \dots, y_n)$. The primary concern then is that the classifier as a function of the state $\tilde Q(y_1, \ldots, y_n \vert s)$ should be as close as possible to the ground-truth $\tilde p(y_1, \ldots, y_n \vert s)$ rather than the absolute value of the classification loss (though we do want the loss the be as low as possible). In the experiments we have considered, there is a theoretical lower bound on the classification loss as the base distributions we have considered have some overlap.
>
> We have collected additional empirical results regarding classifier accuracy and its effect on the constructed composition in the rebuttal PDF. We will also add these results to the paper. Figures R.1, R.2, and R.3 show the cross-entropy loss of the classifier for terminal and non-terminal states (eqs (10) and (12)) as a function of the number of training steps for the GFlowNet 2D grid domain, the molecular generation domain, and the Colored MNIST digits domain respectively. They show that the loss drops quickly but remains above 0. Figure R.1 further shows the distance between the composition and the ground truth as a function of the number of training steps for the classifier. This shows that the distance to the ground truth falls quickly in conjunction with the loss.
>
> > How does the training of the classifier compare to simply training a GFlowNet with the appropriate composition from scratch in terms of runtime?
>
> In addition to the new training curves in figures R.1-R.3, we have added new results regarding the runtime of classifier training in Tables R.1 and R.2 of the rebuttal PDF. These tables show the total runtime, as well as separate measurements for the time spent sampling trajectories and training the classifier.
>
> The runtime for training the classifier is of the same order of magnitude as training the base models. The main computational expense when training the classifier comes from sampling from the base models, comprising 70%-90% of the runtime, rather than training the classifier itself. In this regard there is certainly room for improvement in the training procedure, e.g. by sampling fewer trajectories, by training on individual trajectories multiple times, and by reducing the number of training steps (Figures R.1-R.3 show that loss plateaus quickly in all cases).
>
> However, we would like to stress that training the appropriate composition from scratch is far from trivial, and that our approach is more general. Specifically, in the case of GFlowNets, training the model requires access to the composite reward, which may not be available. Even if the base reward functions ($R_1, R_2$) are available, compositions can’t be expressed through the rewards only because operations on rewards are not analogous to operations on probabilities. For example, $\frac{R_1(x) R_2(x)}{R_1(x) + R_2(x)}$ is not a valid reward for the harmonic mean. In the case of diffusion models, training requires samples from the composition, which are not available before realizing the composition. In addition, once the classifier has been trained, it can be used to sample from all supported compositions of the base models, rather than a single composition as would be the case for a GFlowNet or diffusion model trained from scratch. We will make these points clear in the paper.
>
> > What are the challenges in applying the method to more complicated tasks with diffusion models?
>
> The challenges are primarily practical. More complex diffusion models are more expensive to train and sample from. More expensive sampling also makes training the classifier and sampling from the composition more expensive. More complex compositions involving more base models and more observations require either more classifiers or larger classifiers (with more outputs).
>
> > Is it conceivable to be able to compose a mixture of GFlowNets and diffusion models on mixed continuous/discrete tasks?
>
> This is certainly interesting future work. For the current method, this is not possible as the base models must have a shared domain. However, composing diffusion models with GFlowNets with continuous domain, or GFlowNets with discrete diffusion models is certainly possible.
>
> > Is there a reason to be limited to composing only 3 models?
>
> No; there is no theoretical limit on the number of base models. Generally, we felt that 3 base models is a reasonably realistic setting.
>
> > The main PDF on OpenReview appears to have some old version of the Appendix at the end. I assume that was by mistake?
>
> Yes, this was an oversight on our part. We apologize for any confusion this may have caused. The correct appendix can be found in the supplement.

---

> > ### Comment · Reviewer_eEwT · 2023-08-14
> > **Response to rebuttal**
> >
> > Thank you for your response and apologies for the delayed response!
> >
> > > However, despite the known connection, classifier guidance for GFlowNets has not been proposed in prior work as far as we know.
> >
> > I agree, however I believe it would useful to highlight this connection in the paper.
> >
> > > Figures R.1, R.2, and R.3 show the cross-entropy loss of the classifier
> >
> > Thanks for these additional experiments. As you pointed out later in your rebuttal - there doesn't seem to be much improvement after a few hundred steps which is a bit surprising. Additionally, this makes it hard to (empirically) understand the effect of the classifier performance since we there isn't much range captured by the experiment.
> >
> > > Efficiency of training the classifiers
> >
> > Thanks for these details and the clarification. The training cost of the classifier is similar to training the GFlowNet from scratch. This is something which should be clarified in the paper, along with the computational challenges of using it with larger and more sophisticated diffusion models.
> >
> > I appreciate the authors response which answered most of my questions. I still believe the weaknesses remain, but the paper is strong enough for acceptance.

---

### Official Review · Reviewer_VKR5 · 2023-07-13

**Soundness:** 3 good
**Presentation:** 3 good
**Contribution:** 3 good
**Rating:** 7
**Confidence:** 2

**Summary:**

The paper proposes a method to compose multiple iterative generative models, i.e., either multiple GFlowNets or multiple diffusion models. The idea starts out with a mixture model over the generative models. Then, one can construct a categorical distribution over the generative models that tells us which model a sample originated from. By adapting classifier guidance to GFlowNets, the proposed method can compose multiple models in a way that allows both emphasizing or de-emphasizing specific models by treating the different generative models as different classes. On a diverse set of (toyish) experiments the method is shown to be effective for both GFlowNets and diffusion models.

**Strengths:**

The method is very interesting. In particular, the part where the question "which model was this sampled from" is treated as a classification task for the purpose of compositional generation.

The paper addresses a very important problem with high impact.

The presentation is easy to follow and the text is well-written.



**Weaknesses:**

The experiments clearly demonstrate the effectiveness and versatility of the proposed method. Though, the experiments are limited to toyish settings and I believe more complicated settings would greatly enhance the impact of this work.

**Questions:**

All state-of-the-art diffusion models for images allow for text conditioning. Conditioning on different texts can be viewed as creating multiple different generative models from the viewpoint of this paper. I believe this is a more realistic setting than assuming that one would want to combine multiple diffusion models that were trained entirely separately from one another. It would be interesting to see some evaluations of this setting. A cheap strategy to obtain a classifier for this could be to fine-tune (part of) the CLIP model.

**Limitations:**

The main limitation is inherited from classifier guidance: the need to train a classifier on intermediate states. This is already mentioned in the paper.

---

> ### Author Rebuttal · Authors · 2023-08-09
>
> > The experiments clearly demonstrate the effectiveness and versatility of the proposed method.
>
> Thank you for the positive feedback and insightful comments!
>
> > The experiments are limited to toyish settings and I believe more complicated settings would greatly enhance the impact of this work.
>
> We naturally agree. Model composition with diffusion models and GFlowNets is still new. Our goal for this work was to formulate the problem and motivate further empirical study through our illustrative examples. We consider larger scale problems, such as composing state of the art image generation models to be a natural next step.
>
> > Conditioning on different texts can be viewed as creating multiple different generative models. It would be interesting to see some evaluations of this setting.
>
> We did consider using conditioned models to emulate multiple distributions. One particularly interesting application is in safety, where one could compose safety or moral constraints on an existing text-conditioned generative model to remove harmful content.
>
> That said, we believe studies like this warrant separate comprehensive evaluation. In the current work, we focus on the development of a new approach to model composition, its theoretical foundation, and empirical validation.

---

> ### Comment · Reviewer_VKR5 · 2023-08-15
>
> Thanks to the authors for their response. I believe the points raised by reviewer eEwT and Ean9 on the related work (classifier guidance, compositional generation, etc.) are important to properly highlight in the paper. If these points are addressed I vote for acceptance.

---

### Official Review · Reviewer_ZgyF · 2023-07-26

**Soundness:** 3 good
**Presentation:** 3 good
**Contribution:** 4 excellent
**Rating:** 7
**Confidence:** 4

**Summary:**

The current paper focuses on the challenge of composition generation from pretrained generative models, with a specific focus on GFlowNets and Diffusion models. In comparison to prior literature, two novel compositionality operations are introduced for generating samples that are simultaneously likely according to two generative models or likely per a subset and unlikely per the remaining models. This is a strict generalization of operations introduced in prior work on composition of energy-based models. Practically, the operations are instantiated via a framework motivated by classifier guidance in diffusion models. Experiments are conducted on a molecule generation application and a colored MNIST problem.

Rebuttals acknowledgment: I had a good view of the paper before rebuttals and the authors' response to my questions was fair. I continue to keep my score accordingly.

**Strengths:**

I really like this work! The contributions are simple and straightforward, but very interesting. The formalization, generalized operations, and relation to classifier guidance were exciting to read through. The authors also appropriately acknowledge the limitations of their work, specifically the need for sufficiently strong component models.

**Weaknesses:**

- My biggest apprehension is limited experimental investigation, which would raise the quality of this paper quite strongly in my opinion. I do not hold this to be a strong weakness though.

**Questions:**

As noted in the introduction, use of large scale generative models has been seen in several tasks in the foundation model era of machine learning. Given the limited empirical evaluation, I would like to gather the authors' thoughts on use of their frameworks for, say, controllable generation of language via LLMs or controllable generation of images via text-diffusion models (e.g., what would the model show for known compositionality failures of Dall-e and related methods? See [1]).

[1] Conwell and Ullman, 2022. https://arxiv.org/abs/2208.00005.

**Limitations:**

- A clear note of what is meant by compositionality would help in this work, since the term is extremely overloaded. The experiments currently reported focus on what would be called systematicity or systematic generalization in my opinion (see Hupkes et al., "Compositionality decomposed"), but other valuable forms of compositionality, e.g., productive generalization, will arguably require some "chaining" operator that allows prior generated states to be fed into the model for generating the next state. Since the work focuses on GFlowNets and diffusion models, where a notion of sequential generation of intermediate states is present, arguably authors can use their defined operations to perform productive generalization as well? I would appreciate if the scope of this paper is clearly discussed.

---

> ### Author Rebuttal · Authors · 2023-08-09
>
> Thank you for your thoughtful review. We address specific questions below:
>
> > I would like to gather the authors' thoughts on use of their frameworks for, say, controllable generation of language via LLMs or controllable generation of images via text-diffusion models (e.g., what would the model show for known compositionality failures of Dall-e and related methods?
>
> Thanks for recommending the Conwell *et.al.* paper! The CLIP training objective treats language prompts as a "bags-of-words" for compute reasons. Therefore, it is not surprising that the image generated by DALL-E 2 (and family) lack relational understanding. This type of constraint is the norm, not the exception in large model training, which is why we believe finding better ways to model relationships is an important area to study.
>
> The type of relationships mentioned by Conwell *et.al.* are often mappable to spatial arrangements. It is not difficult to envision learning multiple base models that each model specific relationships. Complex relational queries could then be represented as a composition of appropriate base models, and samples capturing these relations could be generated using the method we have proposed here. This is similar to prior work [27] which used multiple EBMs to model a number of relationships, and found that samples from appropriate compositions of these EBMs reproduced the target relationships significantly more faithfully than StyleGAN2 conditioned on a textual encoding of the relationships. We leave this to follow-up works, as it warrants comprehensive evaluation.
>
> > A clear note of what is meant by compositionality would help. I would appreciate if the scope of this paper is clearly discussed.
>
> Since submission we have worked hard to further clarify our exposition on compositional sculpting and our method. We have clarified that we focus on a narrow but well-defined type of composition where we look to algebraically combine (compose) probability densities in a controllable fashion, such that we can emphasize or de-emphasize regions in the composition where specific base distributions have high density. The harmonic mean and contrast operations we highlight in the paper are specific instance of this. Our paper focuses on a setting where we have access to GFlowNets or diffusion models which can generate samples from those probability distributions we wish to compose.
>
> The iterative nature of GFlowNets and diffusion models is preserved when composed using our method. Thus, if the base models exhibit productive generalization, so will their compositions. In addition, we would like to highlight that compositions themselves can be chained as well. As the compositions correspond to valid GFlowNets or diffusion models, one can compose these compositions with other GFlowNets or diffusion models.

---

> > ### Comment · Reviewer_ZgyF · 2023-08-13
> >
> > Thank you to the authors for their response. I'll keep my score as is.

---

### Official Review · Reviewer_UsMn · 2023-07-27

**Soundness:** 3 good
**Presentation:** 3 good
**Contribution:** 3 good
**Rating:** 6
**Confidence:** 4

**Summary:**

This paper introduces a method to combine sequential generative models, in this case GFlowNets, so as to create new distributions from base models. This is done by training classifiers that are then used to guide sampling. The method is tested on a simple grid and a molecular domain (emulating the problem of the paper that introduced GFlowNet), as well as on MNIST with diffusion.

**Strengths:**

The paper is moderately easy to read, although some of its results were not immediately clear to me so I spent quite some time doodling on paper to convince myself that the propositions were reasonable.

What the paper proposes and seems to be able to achieve empirically is very interesting. Combining generative models in the ways shown here could be an amazing multiplier of large pretrained models.

**Weaknesses:**

Generally the paper is not making a good job of convincing the readers that the proposed method should work at the theory level, and that the effort of combining distributions by training a classifier is worth it (compared to retraining a generative model).

**Questions:**

l211 and around, I’m not sure I understand the move to approximate $p(y_i|s)$ with $p(y_i|x)$, even when $s$ and $x$ are related by $\tau =(s_0,…,s,…x)$ a valid trajectory. How or when are these two quantities interchangeable?
The objective in 12 involves $w_i$, but the paragraph following (11) seems to suggest that $Q(y_2, .., y_n|x)$ can be replaced by the $w_i$s. So I’m confused whether (12) uses the loss described in (11) or a modified version of it. Another worrying aspect of eq (12) is that there’s an $O(nm)$ sum, which sounds like it can get expensive, and there’s a product of $(n-1)$ probabilities, which sounds like it can get awfully numerically unstable. I’m surprised the authors have been able to train models at all. Are there any tricks involved?

The appendix is incomplete. In fact, part of the proposed contribution of the paper is to provide a combination method for diffusion methods. This is never quite explained properly; readers are directed to appendix D which is empty. In addition, although the propositions and theorems are analogs of past work, it was quite surprising to find theorems with no proofs in a paper. Even if the proof is almost identical to prior work, reproducing it with proper credit seems like the least thing to do; in this case there are nuances with the GFlowNet framework that are left unexplained.

The authors already highlight this limitation, but this seems like something quite fundamental that is for some reason not reported: **How expensive is it to train the classifier**? If it’s just as expensive as training a new generative model, then there’s little incentive to use the proposed method. I’d like not to have to take the authors’ word for it and instead see empirical evidence, training curves, wall time, and so on.

Another missing result is some validation that the learned distributions are the expected ones (e.g. a plot showing that the JS divergence goes to 0 with more training/capacity) at least on a toy setting like the grid environment.

I really appreciate what the paper is trying to accomplish, and the empirical results seem very nice (although lacking some crucial results). I’m not sure though that the proposed method is correct, and with no proofs or extra details of why this should work I’m really inclined to reject this paper. Happy to engage in conversation with the authors of course.

**Limitations:**

The authors have addressed some of the limitations of their work.

---

> ### Author Rebuttal · Authors · 2023-08-09
>
> Thank you for the review and feedback on our paper.
>
> > The appendix is incomplete
>
> The main-text PDF included an incomplete draft of the appendix by mistake. We apologize for the accidentally caused confusion. The complete appendix is provided in the supplementary zip-archive (can be downloaded from the openreview page). The proofs of all theoretical results as well as the formulation of the approach for diffusion models are in the supplement.
>
> > How expensive is it to train the classifier?
>
> Following your suggestion, we empirically evaluated classifier training time and learning curves. The results are shown in Figures R.1, R.2, R.3, and Tables R.1, R.2 in the rebuttal PDF.
>
> The classifier training time is comparable to the base generative model training time. However, most of the classifier training time (more than 70%, or even 90%) was spent on sampling trajectories from the base generative models. Our implementation of the training could be improved in this regard, e.g. by sampling a smaller set of trajectories once and re-using trajectories for training and by reducing the number of training steps (the loss curves show that classification losses plateau quickly).
>
> We would like to note that the composite distributions can not be realized by training new generative models for the target composition directly. For diffusion models, training requires data from the target distribution, which is not available without realizing the composition first. For GFlowNets, training the model requires access to the composite reward (which might not be available). Even if the base reward functions ($R_1,R_2$) are available, the compositions (i.e. harmonic mean) can’t be expressed through the rewards only, i.e. $\frac{R_1(x)R_2(x)}{R_1(x)+R_2(x)}$ is not a valid reward for the harmonic mean, since rewards are unnormalized.
>
> We also note, that after the classifier is trained once, it can be used to construct multiple compositions of the base distributions.
>
> > Validation that learned distributions are the expected ones
>
> Figure R.1 (right) in the rebuttal PDF, shows the evolution of the distances between learned compositions (realized via classifier) and the ground-truth composition distributions (in 2D grid domain). For all compositions, as the classifier training progresses, the distance to the ground truth distribution decreases. Compared to the distance at initialization we observe almost an order of magnitude distance reduction by the end of the training.
>
> > l211 and around, I’m not sure I understand the move to approximate $p(y_i| s)$ with $p(y_i|x)$. How or when are these two quantities interchangeable?
>
> We would like to clarify that we do not propose to approximate $p(y_i|s)$ with $p(y_i|x)$. The objective in eq. (12) uses $\ell(\hat\tau,\hat y_1,\ldots,\hat y_n;\phi)$. The value of $\ell$ appearing in (12) is exactly the same as given in eq. (11).  We arrive at eq. (12) by combining several ideas.
>
> 1. Our goal is to train a classifier $\tilde Q(y_1,\dots,y_n|s)$. This classifier can be obtained as the optimal solution of
> $\min\limits_\phi \operatorname*{\mathbb{E}}\limits_{\hat\tau,\hat y_1,\dots,\hat y_n\sim\tilde p(\tau,y_1,\dots,y_n)}\ell(\hat\tau,\hat y_1,\dots,\hat y_n;\phi),$
> where $\ell$ is defined in eq. (11). We can obtain an unbiased estimate of the loss (and its gradient) by sampling $(\hat\tau,\hat y_1,\dots,\hat y_n)$ and evaluating (11) directly. The challenge is not in computing (11), but in the expectation over $(\tau,y_1,\dots,y_n)$. The steps described in the paragraphs following (11) were introduced to obtain an estimate of this expectation.
>
> 2. The expectation above can be expressed as
> $\operatorname*{\mathbb{E}}\limits_{\hat\tau, \hat y_1\sim\tilde p(\tau,y_1)}\left[\sum\limits\_{\hat y_2=1}^m\dots\sum\limits\_{\hat y_n=1}^m \left(\prod\limits\_{i=2}^n \tilde p(y_i=\hat y_i|x=\hat x)\right)\ell(\hat\tau,\hat y_1,\dots,\hat y_n;\phi)\right],$
> where we re-wrote the expectation over $(y_2,\dots,y_n) | \tau$ in the from “expectation” = sum(“probability” * “value”). The expectation over $(\tau,y_1)$ can be estimated by sampling pairs $(\hat\tau,\hat y_1)$ as described in the paragraph after eq. (11). The only missing part is the probabilities $\tilde p(y_i=\hat y_i|x=\hat x)$ which are not directly available.
>
> 3. Our proposal is to approximate these probabilities as $\tilde p(y_1=j|x=\hat x)\approx w_j(\hat x;\phi)=\tilde Q_\phi(y_1=j|x=\hat x)$. The idea here is that the terminal state classifier $\tilde Q_\phi(y_1|x)$, when trained to optimality, produces outputs exactly equal to the probabilities $\tilde p(y_1|x)$.
>
> 4. Steps 1-3, give a procedure where the computation of the non-terminal state classification loss requires access to the terminal state classifier. As we described in the paragraph preceding eq. (12), we propose to train non-terminal and terminal classifiers simultaneously and introduce “target network” parameters. The weights $w$ are computed by the target network $\tilde Q_{\bar\phi}$.
>
> > Another worrying aspect of eq (12) is that there’s an $O(nm)$ sum, which sounds like it can get expensive
>
> In our experiments $n$ and $m$ were at most $3$, and the computational cost of summation over $\hat y_2,\dots,\hat y_n$ was small. In general, one could trade off estimation accuracy for improved speed by replacing the summation with Monte-Carlo estimation. In this case, the values $\hat y$ are sampled from the categorical distributions $Q_{\bar \phi}(y|x)$. Note that labels can be sampled in parallel since $y_i$ are independent given $x$.
>
> > There’s a product of $(n-1)$ probabilities, which sounds like it can get awfully numerically unstable. I’m surprised the authors have been able to train models at all. Are there any tricks involved?
>
> We only employed the standard techniques for improving the numerical stability of operations on probabilities (re-parameterization in log-probabilities) and did not observe any numerical instability issues.

---

> > ### Comment · Reviewer_UsMn · 2023-08-16
> >
> > Thanks for all the precisions. I'm still having a hard time internalising why this works, but unfortunately do not have the time to dig into it. I will raise my score since you've addressed my concerns.
> >
> > > The classifier training time is comparable to the base generative model training time.
> >
> > It does seem like there's room for improvement here, but the loss does seem to plateau pretty fast on the classifier. Maybe this is somewhere where scale will make the gap clearer (intuitively it should be harder to train the generator, but science is all about beating intuitions so...)
> >
> > > re-parameterization in log-probabilities
> >
> > That makes sense; maybe worth mentioning (this apparently is a pretty central trick to making GFlowNets work as well).

---

### Author Rebuttal · Authors · 2023-08-09

We thank all reviewers for the time and effort dedicated to review of our work and for the helpful and constructive feedback.

## Motivation and focus of the paper

Our work is motivated by the growing costs of general-purpose pre-training of generative models as well as the need for model reuse and control of the generation post-training. The problem of model composition is still new and the iterative nature of the generation processes in diffusion models and GFlowNets necessitates special methods.

We have developed a formal approach where composition is defined as meaningful mathematical operations on a set of base probability distributions. These compositions are highly controllable, allowing us to emphasize or de-emphasize regions in the composition where specific base distributions have high density. We assume that we have access to a number of GFlowNets or diffusion models that generate samples from these base distributions, and provide a method to construct processes that generate samples from the composite distributions. Compared to training GFlowNets or diffusion models from scratch to reproduce these compositions, which is generally impractical or impossible, our method is both practical and more flexible, as after training it can generate samples from all supported compositions. Further, we derived generalized variants of the theoretical results on diffusion mixture and guidance for the case of GFlowNets. Following theoretical justification, we empirically validate the approach in a range of experimental settings including a practically-relevant molecule generation task.

We believe that our work will motivate and support future research on principled approaches to generative model composition with potential for scale.

## Additional experimental results

We have collected additional empirical results to support our response to reviewers’ comments. The new figures and tables are in the rebuttal PDF document (attached to this comment). We list the new results below, and discuss them in detail within the individual responses to reviewers.

### Classifier training time. Analysis of classifier and learned distributions.

Following the suggestions made by reviewers UsMN and eEwT, we provide classifier training curves (Figures R.1, R.2, R.3) and a summary of the training time (Tables R.1, R.2).  Figure R.1 also shows the distance between the learned compositions (obtained via our method) and ground-truth compositions in the 2D GFlowNet domain.

### Effect of reward exponent $\beta$ on the GFlowNet models: mode coverage and diversity

Following reviewer Eean9’s suggestions, we have evaluated the sample diversity of the base GFlowNets in the molecule generation domain at different reward exponents $\beta. The results are shown in Tables R.3 and R.4.

## Clarity improvement and subsequent revision
Based on the feedback from the reviewers, we will expand the paper and incorporate a number of clarifications. We list the most important changes below:

- We will add the additional experimental results (presented in the rebuttal PDF) and the corresponding discussion.
- Following reviewer UsMn’s suggestion, we will elaborate on the details of the derivation of the non-terminal state classification loss (12), as well as computational complexity and numerical stability.
- Following reviewer ZgyF’s suggestion, we will extend the discussion of the application of the method for controllable text generation and image generation via text-diffusion models.
- Following reviewer ZgyF’s suggestion, we will discuss the scope of the paper and clarify the notion of compositionality in the context of our work as well as the relation to other forms of compositionality.
- Following reviewer VKR5’s suggestion, we will extend the discussion of the application of the method for the composition of models obtained by conditioning on different text prompts in the context of text-conditioned diffusion models.
- Following the suggestions of reviewers eEwT and Ean9, we will expand the discussion of the related work on composition and guidance in generative models as well as unifying connections between GFlowNets and diffusion models.
- Following reviewer Ean9’s suggestion, we will add a subclaim explaining why the classifier-guided policy (8) is valid (sums up to 1 over $s’$) and explain the details of the practical implementation of the policy.
- Following reviewer m7vZ’s suggestion, we will add “Broader impact” section.

---

### Decision · Program_Chairs · 2023-09-21

**Decision:**

Accept (poster)

**Comment:**

This paper proposes the composition of pretrained generative component models to
 achieve compositional generalization, and it's applicable to iterative generative processes such as GFlowNets and diffusion models.
The main contribution is the introduction of two new composition operators: harmonic mean and contrast.
After rebuttal, the authors addressed most of the concerns on missing details and extra experiments.
Six reviewers reviewed this paper and all recommended acceptance.
Based on the reviewers' feedback, the decision is to recommend the paper for acceptance.